# Long-Tailed Distribution-Aware Router For Mixture-of-Experts in Large Vision-Language Model

## Abstract

The mixture-of-experts (MoE) architecture, which replaces dense architectures with sparse ones, has garnered attention in large vision-language models (LVLMs) for achieving comparable performance with fewer activated parameters. Existing MoE architectures for LVLMs primarily focus on token-to-expert routing (TER), encouraging different experts to specialize in processing specific tokens. However, these architectures typically rely on the load balancing mechanism, neglecting the inherent distributional differences between vision and language modalities. To address this, we propose the **L**ong-**T**ailed **D**istribution-aware **R**outer (LTDR) for vision-language TER, which tackles two key challenges: (1) Modality-specific distribution-aware routing. We observe that language TER follows a relatively uniform distribution, whereas vision TER exhibits a long-tailed distribution. This modality discrepancy necessitates specific routing strategies for each modality. (2) Vision-specific dynamic expert activation. Recognizing the importance of high-information vision tail tokens, we introduce an oversampling-like strategy by increasing the number of activated experts to sufficiently learn vision tail token representations. Experiments on extensive vision-language and vision benchmarks validate the effectiveness of our approach.

## 1 Introduction

Recent advances in large vision-language models (LVLMs) Achiam et al. (2023); Dubey et al. (2024), which bridge the vision-language gap, have demonstrated impressive instruction-following and generalization capabilities. However, real-world applications necessitate models that can handle diverse tasks. Traditional approaches that train separate models for each task incur significant redundancy and resource consumption. Despite efforts to expand datasets or models Chen et al. (2023b); Bai et al. (2023c); Lu et al. (2024), the demand for substantial resources remains.

The mixture-of-experts (MoE) Jacobs et al. (1991) architecture enables scalable parameter growth without a proportional increase in inference costs. Its effectiveness in model scaling has been demonstrated in various recent works Dou et al. (2023); Gou et al. (2023); Bai et al. (2023a); Chen et al. (2024b); Dai et al. (2024). Specifically, MoE-LLaVA Lin et al. (2024a) achieves performance comparable to LLaVA-7B and LLaVA-13B Liu et al. (2023a) while activating only 3B parameters.

The core of MoE lies in its token-to-expert routing (TER). While most implementations Shazeer et al. (2017) utilize trainable routers to predict routing probabilities, they typically enforce load balancing constraints on TER to prevent expert overload or underload, thereby promoting a uniform distribution of tokens across experts. However, this uniform-load strategy is sub-optimal for multi-modal tasks, where vision tokens follow a long-tailed distribution He et al. (2016); Krizhevsky et al. (2017) in contrast to the more uniform distribution of language tokens Vaswani et al. (2017); Devlin et al. (2019). Indiscriminate load balancing hinders the effective learning of vision tokens. As illustrated in Fig. 1 (a), vision tokens comprise a majority of low-information background tokens (the head) and a minority of high-information foreground tokens (the tail). Enforcing load balancing scatters these critical, but sparse, foreground tokens across different experts, which impedes the consolidation of similar information and prevents the selection of specialized experts for these important tokens.

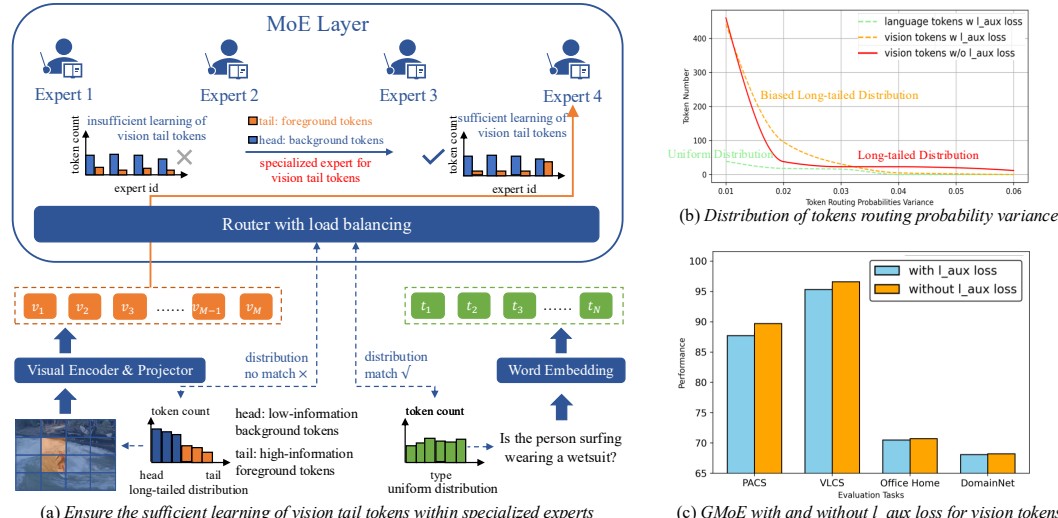

(a) *Ensure the sufficient learning of vision tail tokens within specialized experts*

(b) *Distribution of tokens routing probability variance*

(c) *GMoE with and without l_aux loss for vision tokens*

Figure 1: (a) Our goal is to ensure that sparse yet high-information vision tail tokens are sufficiently learned within specialized experts. (b) Distribution of TER probability variance. While Language TER with load balancing is uniform, vision TER without load balancing exhibits a long-tailed distribution and vision TER with load balancing shows a biased long-tailed characteristic. (c) GMoE with and without load balancing. Removing load balancing from vision tokens improves performance.

We perform an analysis of this question. Fig. 1(b) plots the TER probability variance (x-axis) against the token count (y-axis). The distribution of vision token counts across probability variances is long-tailed. The majority of tokens, which possess low probability variances, typically correspond to low-information background tokens, as the router struggles to assign specialized experts for them. In contrast, high-information foreground tokens generally exhibit high variances. As shown in Fig. 1(b), the application of load balancing reduces the count of tokens with high variances, suggesting that the load balancing hinders the expert specialization for foreground tokens. While language token counts are uniformly distributed across probability variances, indicating that language tokens are compatible with load balancing. Fig. 1(c) shows that removing load balancing for vision tokens leads to improved performance. Based on these observations, our goal is to ensure that sparse yet high-information vision tail tokens are sufficiently learned within specialized experts as shown in Fig. 1(a).

To ensure sufficient learning of vision tail tokens, we propose the **L**ong-**T**ailed **D**istribution-aware **R**outer (LTDR). Our approach solves two key challenges: (1) *Modality-specific distribution-aware routing.* Increasing the variance of routing probabilities allows vision tokens, especially vision tail tokens, to be learned within specialized experts. Consequently, we remove load balancing for vision TER to align with its long-tailed distribution, thus improving the variance of routing probabilities. For language TER, we retain load balancing as it aligns well with its uniform distribution. (2) *Vision-specific dynamic expert activation.* Recognizing the importance of high-information vision tail tokens, we employ an oversampling-like strategy to increase the number of activated experts for them, thereby improving fault tolerance during specialized expert selection. Experiments on vision-language and vision benchmarks confirm that our approach significantly enhances visual understanding.

Our contributions can be summarized as follows:

- We identify the detrimental effect of traditional load balancing on vision tokens. This effect stems from a distribution mismatch, where the uniform routing preference of load balancing leads to insufficient learning of important, yet long-tailed, vision tail tokens.
- We propose the long-tailed distribution-aware router (LTDR), which contains two key modules: (1) Modality-specific distribution-aware router for vision-language tasks with diverse distributions; and (2) Vision-specific dynamic expert activation for vision tail tokens that necessitate sufficient representation learning.
- Extensive experiments demonstrate the effectiveness of our approach, achieving average improvements of 1.2%, 2.0% and 1.0% on benchmarks, providing new insights for MoE.

## 2 RELATED WORKS

### 2.1 LARGE VISION-LANGUAGE MODELS

The success of large language models (LLMs) has accelerated the development of LVLMs Lin et al. (2024a). GPT-4 Achiam et al. (2023) and LLaVA Liu et al. (2023a) encode visual inputs into LLMs-compatible representations, enabling seamless vision-language integration. Current research focuses on dataset scaling Zhang et al. (2023) and parameter-efficient adaptation Houlsby et al. (2019); Lester et al. (2021); Hu et al. (2021); Ye et al. (2023) to enhance efficiency and scalability. However, LVLMs still face critical challenges, including limited task generalization and rapidly escalating inference costs with parameter growth. To this end, MoE has emerged as a promising solution.

### 2.2 LONG-TAILED DISTRIBUTION

Real-world exhibits a long-tailed data distribution, where a few classes contain abundant samples while most classes suffer from severe data scarcity Liu et al. (2019); Cui et al. (2019); Kang et al. (2020); Menon et al. (2020). This class imbalance significantly impairs deep learning model training. Current approaches typically fall into resampling Han et al. (2005); Buda et al. (2018), reweighting Lin et al. (2017); Wang et al. (2017), and logits adjustment Ren et al. (2020); Hong et al. (2021). On vision-language tasks, the challenge intensifies as the long-tailed distribution manifests at both samples and tokens, creating difficulties for effective model training and optimal performance.

### 2.3 MIXTURE-OF-EXPERTS

MoE usually uses a trainable linear layer for routing probability prediction Lepikhin et al. (2020). Task routing Gururangan et al. (2021); Jain et al. (2024); Zhou et al. (2024) assigns task-specific tokens to predetermined experts, while cluster routing Dou et al. (2023); Gou et al. (2023) groups feature-similar tokens into the same expert. Dynamic routing Huang et al. (2024); Guo et al. (2024) adaptively adjusts expert numbers to reduce hyper-parameter constraints. Recent advances have introduced more sophisticated techniques, including task relationship modeling Ma et al. (2018), gradient conflicts mitigating Yang et al. (2024), and sequential routing Zhong et al. (2024). Nevertheless, they neglect the issue of routing under long-tailed distribution, which are normal in practical applications. To our best knowledge, Wang et al. (2020); Jin et al. (2023) focus on sample-level long-tailed distribution. However, they only use traditional routing strategies to mitigate model bias, without introducing new routing strategies. Our LTDR is dedicated to addressing the long-tailed distribution situation at token-level . Moreover, LTDR can process data from diverse modalities through distinct routing strategies, thereby offering a more comprehensive solution to this complex problem.

## 3 METHODOLOGY

### 3.1 PRELIMINARIES

**Large Vision-Language Models.** LVLMs integrate the capabilities of LLMs with visual processing technologies, enabling vision-language understanding and generation. As shown in Fig. 1 (a), the text input is first transformed through a word embedding, which projects the text $\mathbf{t}$ into a continuous vector space, resulting in the language token sequence $\mathcal{T} = [t_1, t_2, \cdots, t_N] \in \mathbb{R}^{N \times D}$. $N$ means the sequence length of language tokens, and $D$ denotes the hidden layer size of the LLM.

Similarly, the RGB image input $\mathbf{v} \in \mathbb{R}^{H \times W \times 3}$, where $H$, $W$, and 3 denote the height, width, and channels of the image at its original resolution, the visual encoder processes the image to extract a sequence of vision tokens $\mathcal{Z} = [z_1, z_2, \cdots, z_M] \in \mathbb{R}^{M \times C}$. $M$ is the sequence length of vision tokens, and $C$ is the hidden layer size of the visual encoder. To align vision tokens with language tokens in the same vector space, a visual projection layer is employed to map $\mathcal{Z} \in \mathbb{R}^{M \times C}$ to $\mathcal{V} = [v_1, v_2, \cdots, v_M] \in \mathbb{R}^{M \times D}$, where $D$ matches the hidden layer size of the LLM. This alignment ensures that both vision and language can be processed jointly by the subsequent layers of the model.

Subsequently, vision and language tokens are concatenated and fed into the LLM. The LLM consists of layers of multi-head self-attention (MSA) and feed-forward network (FFN), with layer normalization (LN) and residual connection applied to stabilize training and enhance performance. As shown in

Eq. 1 $\sim$ Eq. 4, where $L$ is the layer number of LLM, the LVLM achieve a deep understanding of the relationships between vision and language, enabling effective performance on vision-language tasks.

$$\mathbf{x}_0 = [v_1, v_2, \cdots, v_M, \cdots, t_1, t_2, \cdots, t_N], \tag{1}$$

$$\mathbf{x}'_\ell = \text{MSA}(\text{LN}(\mathbf{x}_{\ell-1})) + \mathbf{x}_{\ell-1}, \ell \in \{1, \ldots, L\}, \tag{2}$$

$$\mathbf{x}_\ell = \text{FFN}(\text{LN}(\mathbf{x}'_\ell)) + \mathbf{x}'_\ell, \ell \in \{1, \ldots, L\}, \tag{3}$$

$$\mathcal{Y} = \text{LN}(\mathbf{x}_L), \tag{4}$$

The output of the LVLM is optimized through a generative loss in an auto-regressive manner. Given an image and its corresponding instruction text, the LVLM aims to generate the output text sequence $\mathcal{Y} = [y_1, y_2, \cdots, y_O] \in \mathbb{R}^{O \times D}$ by progressively prediction, where $O$ is the sequence length of the text output. The loss function is defined in Eq. 5, $\mathcal{Y}_{<i}$ indicates the output sequence before token $y_i$, $\theta$ denotes the trainable parameters of the model. We only calculate the loss for the generated text.

$$\mathcal{L}_{\text{regressive}} = -\sum_{i=1}^{O} \log p(y_i \mid \mathcal{V}, \mathcal{T}, \mathcal{Y}_{<i}, \theta), \tag{5}$$

**Mixture-of-Experts.** We replace FFN layers as MoE layers, following MoE-LLaVA Lin et al. (2024a). A MoE layer typically contains multiple FFNs, denoted as an experts ensemble $\mathcal{E} = [e_1, e_2, \cdots, e_K]$, $K$ is the number of total experts. The router implements a linear layer to predict the routing probability of assigning tokens to experts. As shown in Eq. 6, the router produces weight logits $f(\mathbf{x}) = \mathbf{W} \cdot \mathbf{x}$, which are normalized by the softmax function. The matrix $\mathbf{W} \in \mathbb{R}^{D \times K}$ denotes the lightweight trainable parameters for routing, and $\mathcal{P}(\mathbf{x})_i$ is the routing score of the input $\mathbf{x}$ for the $i$-th expert. The final output in Eq. 7 is computed as a weighted sum of the outputs from the Top-$k$ experts with the highest softmax probabilities. $\mathcal{E}(\mathbf{x})_i$ is the output of the $i$-th expert, and the weight for each expert is determined by its routing score.

$$\mathcal{P}(\mathbf{x})_i = \frac{e^{f(\mathbf{x})_i}}{\sum_{j=1}^{K} e^{f(\mathbf{x})_j}}, \tag{6}$$

$$\text{MoE}(\mathbf{x}) = \sum_{i=1}^{k} \mathcal{P}(\mathbf{x})_i \cdot \mathcal{E}(\mathbf{x})_i, \tag{7}$$

Due to the presence of multiple experts, it is necessary to impose the expert load balancing constraint on MoE layers. Traditional methods Lin et al. (2024a); Yang et al. (2024) incorporate differentiable load balancing loss Fedus et al. (2022) into each MoE layer to encourage experts to handle tokens in a balanced manner. As shown in Eq. 8, $\mathcal{F}_i$ is the fraction of tokens processed by expert $\mathcal{E}_i$, and $\mathcal{G}_i$ is the average routing probability of expert $\mathcal{E}_i$.

$$\mathcal{L}_{\text{balancing}} = K \cdot \sum_{i=1}^{K} \mathcal{F}_i \cdot \mathcal{G}_i, \tag{8}$$

**Our Method LTDR.** Since language tokens follow a uniform distribution, while vision tokens exhibit a long-tailed distribution, we focus on optimizing vision-language TER to make experts handle different distributional modality tokens effectively. We find that the load balancing mechanism leads to the scattered vision tail tokens in experts, impeding the learning of specialized experts. Therefore, as illustrated in Fig. 2, our method consists of two modules: (1) Modality-specific Distribution-aware Router (MsDaR). We retain load balancing for language TER as it aligns with the uniform distribution of language tokens, while abandon load balancing for vision TER to adaptively align with the long-tailed distribution of visual tokens. Without load balancing, vision tokens, especially vision tail tokens, exhibit higher routing probability variance, enabling expert specialization. (2) Vision-specific Dynamic Expert Activation (VsDEA). Given the high importance of vision tail tokens, we define the head and tail tokens of vision, and increase the number of activated experts for vision tail tokens, achieving an oversample-like strategy to improve fault tolerance and learning effectiveness.

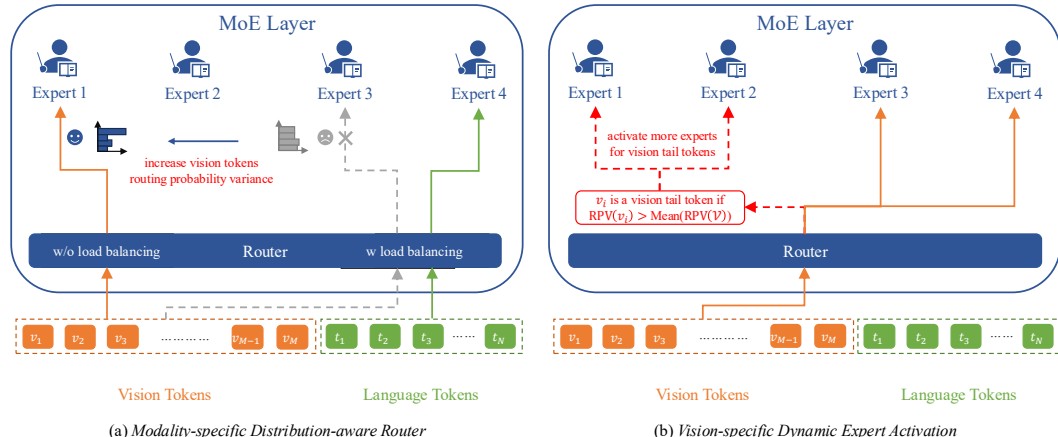

(a) *Modality-specific Distribution-aware Router*   (b) *Vision-specific Dynamic Expert Activation*

Figure 2: (a) We use a modality-specific distribution-aware router, allowing vision and language to be routed with different expert load to adapt to their respective modality distributions. (b) We define the head and tail tokens of vision, and use a vision-specific dynamic expert activation for vision tokens, enabling an oversample-like strategy to make experts process important vision tail tokens sufficiently.

## 3.2 MODALITY-SPECIFIC DISTRIBUTION-AWARE ROUTER

Existing MoE architectures on modality differences fall into modality-aware Nguyen et al. (2024); Chen et al. (2024a); Lin et al. (2024b) and distribution-aware Wang et al. (2020); Jin et al. (2023). Nguyen et al. (2024); Lin et al. (2024b) center on modalities by using a hierarchical MoE and modality-specific expert groups, while Wang et al. (2020); Jin et al. (2023) focus on long-tailed distribution by enhancing expert diversity and reducing dynamic routing. These methods are constrained by load balancing (Eq. 8), without considering modality token distribution differences.

He et al. (2016); Krizhevsky et al. (2017); Vaswani et al. (2017); Devlin et al. (2019) has shown that language follows a uniform distribution, whereas vision exhibits a long-tailed distribution. This divergence stems from the characteristics of vision, which contains a few foreground patches and a large number of background patches. Radford et al. (2021); Kim et al. (2021) also emphasize that the structural and semantic differences between vision and language necessitate specific processing. To this end, we optimize the vision-language TER distribution (Eq. 6) for vision and language.

Load balancing conflicts with the long-tailed vision distribution, an intuitive way is to release vision tokens from load balancing to increase their routing probability variance (RPV). As shown in Eq. 9, for a vision token $v_i \in \mathcal{V}$, its routing probabilities $\mathcal{P}(v_i) \in \mathbb{R}^K$, and $\mathrm{RPV}(v_i)$ is its variance of $\mathcal{P}(v_i)$. The number of vision head tokens is sufficient to generalize across various experts, resulting in uniform routing probability (low RPV). If the limited number of vision tail tokens are distributed uniformly across experts, it would lead to poor generalization. Therefore, a modality-specific distribution-aware router is essential to benefit the sufficient learning of experts for vision tokens. We modify the traditional $\mathcal{L}_{\text{balancing}}$ in Eq. 10, where $\mathcal{T}$ means the language token sequence.

$$\mathrm{RPV}(v_i) = \mathrm{Variance}(\mathcal{P}(v_i)), \tag{9}$$

$$\mathcal{L}_{\text{balancing}} = \sum_{i=1}^{K} \mathcal{F}_i(\mathcal{T}) \cdot \mathcal{G}_i(\mathcal{T}) \tag{10}$$

In this architecture, language tokens maintain their original load balancing, while vision tokens are released from the constraint of load balancing. This enables vision tokens, especially vision tail tokens, which represent information-rich content, to undergo specialized expert processing. As shown in Fig. 2 (a), by enhancing the RPV of vision tokens, these tokens can be allocated to specialized experts instead of being uniformly distributed, facilitating more specialized and efficient learning.

## 3.3 VISION-SPECIFIC DYNAMIC EXPERT ACTIVATION

Given the high importance of vision tail tokens, we seek to enhance their processing for sufficient expert learning. However, the complexity and training instability introduced by long-tailed tokens necessitate a simple yet effective solution. Since the input and output token sequence lengths fix before and after each FFN layer, traditional undersampling or oversampling methods are impractical for adjusting head or tail token counts. To address this, we employ an oversample-like strategy to enhance the expert learning and processing for vision tail tokens without introducing new tokens.

As illustrated in Fig. 2 (b), our innovation lies in defining and identifying vision tail tokens and activating more experts to learn and process vision tail tokens. First of all, we define whether a vision token $v_i \in \mathcal{V}$ is a vision head token or vision tail token by the following Eq. 11.

$$\text{Type}(v_i) = \text{tail}, \text{if } \text{RPV}(v_i) > \text{Mean}(\text{RPV}(\mathcal{V})) \tag{11}$$

Where $\text{Mean}(\text{RPV}(\mathcal{V}))$ represents the mean value of $\text{RPV}(\mathcal{V}) \in \mathbb{R}^M$. We serve vision tokens with larger RPV than the $\text{Mean}(\text{RPV}(\mathcal{V}))$ of total vision tokens as vision tail tokens, and the remainder are considered as heads. This criterion is based on that RPV reflects the TER distribution. Vision tail tokens generally exhibit higher RPV than head tokens, as they are processed by specialized experts. We use mean RPV as the dynamic threshold for vision tail token recognition, as it filters out most of vision head tokens. Subsequently, we route identified vision tail tokens to more experts than originally assigned, resulting in a modified version of Eq. 7 tailored for vision tail tokens in Eq. 12:

$$\text{MoE}(x) = \begin{cases} \sum_{i=1}^{a} \frac{e^{\mathcal{P}(x)_i}}{\sum_{j=1}^{a} e^{\mathcal{P}(x)_j}} \cdot \mathcal{E}(x)_i, & \text{vision tail token} \\ \sum_{i=1}^{k} \frac{e^{\mathcal{P}(x)_i}}{\sum_{j=1}^{k} e^{\mathcal{P}(x)_j}} \cdot \mathcal{E}(x)_i, & \text{vision head \& language token} \end{cases} \tag{12}$$

The weights of the selected experts will be renormalized. Vision tail tokens can be handled by more experts ($k < a \leq K$). Reducing the possibility of expert incorrect routing.

## 4 EXPERIMENTS

### 4.1 EXPERIMENTAL SETUPS

**Benchmarks.** We aim to study the vision TER within the MoE architecture. To ensure a robust evaluation, we conduct experiments on a suite of both vision-language and vision tasks. A detailed description of the benchmarks and tuning datasets can be found in A.1 ∼ A.2. This comprehensive evaluation framework allows for a rigorous assessment of our approach across a diverse capabilities.

**Baselines.** We employ MoE-LLaVA Lin et al. (2024a) and Molmo Deitke et al. (2024) for vision-language tasks, and GMoE Li et al. (2022) for vision tasks. MoE-LLaVA selects the two most relevant experts from a total of four experts, while Molmo selects 8 experts from 64 experts. GMoE is designed for visual generalization and enhances the ability of cross-domain data generalization.

**Configurations.** The experimental setups are as follows. For vision-language tasks, we build upon two base architectures: MoE-LLaVA, evaluated with StableLM-1.6B, Qwen-1.8B, and Phi-2-2.7B backbones, and Molmo, evaluated with OLMo-1B-7B backbone. For vision tasks, we adopt the pre-trained ViT-S/16 backbone following GMoE. Besides, our routing strategy is applied at each training batch. Further implementation details are provided in A.3.

### 4.2 COMPREHENSIVE EVALUATIONS

**Vision-Language Evaluations.** As shown in Tab. 1 ∼ Tab. 2, our method demonstrates robust image-text understanding, achieving superior performance compared to the MoE-LLaVA and Molmo. Specifically, our method achieves an average improvement of 1.2% over StableLM-1.6B, 0.4% over Qwen-1.8B, 0.5% over Phi2-2.7B, and a significant 2.0% improvement over OLMoE-1B-7B. These results substantiate the effectiveness of our method. Furthermore, the consistent improvements observed across diverse model scales highlight the scalability and general adaptability of our method.

Table 1: **Comparison of different LVLMs and our method.** "I", "L", "V", "Q", "P", "M" and "S" respectively represent IDEFICS Laurençon et al. (2023), LLaMA Touvron et al. (2023), Vicuna Chiang et al. (2023), Qwen Bai et al. (2023b), Phi-2 Javaheripi et al. (2023), MobileLLaMA Chu et al. (2023) and StableLM Bellagente et al. (2024). Evaluation benchmarks include GQA Hudson & Manning (2019); SQA[I]: ScienceQA-IMG Lu et al. (2022); VQA[T]: TextVQA Singh et al. (2019); POPE Li et al. (2023); MME Fu et al. (2023); MMB: MMBench Liu et al. (2023b) and MM-Vet Yu et al. (2023). * indicates that there is some overlap in the training data. Sparse models use the configure 4Top2. We calculate the average performance "Avg" across all datasets except for MME.

| Method | LLM | GQA | SQA[I] | VQA[T] | POPE | MME | MMB | MM-Vet | Avg |
|---|---|---|---|---|---|---|---|---|---|
| *Dense Model* | | | | | | | | | |
| I-80B Laurençon et al. (2023) | I-65B | 45.2 | - | 30.9 | - | - | 54.5 | - | - |
| LLaVA-1.5 Liu et al. (2023a) | L-13B | 63.3* | 71.6 | 61.3 | 85.9 | 1531.3 | 67.7 | 35.4 | 64.2 |
| LLaVA-1.5 Liu et al. (2023a) | V-7B | 62.0* | 66.8 | 58.2 | 85.9 | 1510.7 | 64.3 | 30.5 | 61.3 |
| Qwen-VL Bai et al. (2023c) | Q-7B | 59.3* | 67.1 | 63.8 | - | - | 38.2 | - | - |
| TinyGPT-V Yuan et al. (2023) | P-2.7B | 33.6* | - | - | - | - | - | - | - |
| MobileVLM Chu et al. (2023) | M-2.7B | 59.0* | 61.0 | 47.5 | 84.9 | 1288.9 | 59.6 | - | - |
| LLaVA-Phi Zhu et al. (2024) | P-2.7B | - | 68.4 | 48.6 | 85.0 | 1335.1 | 59.8 | 28.9 | - |
| *Sparse Model* | | | | | | | | | |
| MoE-LLaVA-4Top2 | S-1.6B | 60.3* | 62.6 | 50.1 | 85.7 | 1318.2 | 60.2 | 26.9 | 57.6 |
| Our Method | S-1.6B | 61.1* | 63.4 | 51.1 | 86.6 | 1363.5 | 60.6 | 29.9 | 58.8 |
| MoE-LLaVA-4Top2 | Q-1.8B | 61.5* | 63.1 | 48.0 | 87.0 | 1291.6 | 59.7 | 25.3 | 57.4 |
| Our Method | Q-1.8B | 61.6* | 62.8 | 48.9 | 87.2 | 1334.2 | 60.5 | 25.5 | 57.8 |
| MoE-LLaVA-4Top2 | P-2.7B | 61.4* | 68.5 | 51.4 | 86.3 | 1423.0 | 65.2 | 34.3 | 61.2 |
| Our Method | P-2.7B | 62.2* | 68.5 | 52.0 | 86.7 | 1440.8 | 66.7 | 34.0 | 61.7 |

Table 2: **Comparison of Molmo and our method.** Evaluation benchmarks include ChartQA Masry et al. (2022), DocCQA Mathew et al. (2021), AI2D Kembhavi et al. (2016), VQA Goyal et al. (2017), AndroidControl Liu et al. (2024) abd CountBenchQa Beyer et al. (2024). Molmo uses 64Top8 while our method with the VsDEA module uses 64Top12. We calculate the average performance "Avg".

| Method | ChartQA | DocVQA | AI2D | VQA | AndroidControl | CountBenchQA | Avg |
|---|---|---|---|---|---|---|---|
| MolmoE-1B-7B | 65.7 | 79.8 | 85.2 | 82.6 | 81.8 | 74.0 | 78.2 |
| Our Method | 68.1 | 81.3 | 87.4 | 83.3 | 83.2 | 77.6 | 80.2 |

Table 3: **Comparison of GMoE and our method.** Evaluation benchmarks include PACS Li et al. (2017), VLCS Albuquerque et al. (2019), Office-Home Venkateswara et al. (2017) and Domain-Net Peng et al. (2019). We calculate the average performance "Avg".

| Method | PACS | | | VLCS | | | Office-Home | | | DomainNet | | | Avg |
|---|---|---|---|---|---|---|---|---|---|---|---|---|---|
| | 4Top1 | 4Top2 | 6Top2 | 4Top1 | 4Top2 | 6Top2 | 4Top1 | 4Top2 | 6Top2 | 4Top1 | 4Top2 | 6Top2 | |
| GMoE | 86.7 | 88.9 | 87.6 | 96.8 | 92.5 | 96.5 | 70.1 | 70.5 | 71.0 | 67.6 | 68.5 | 68.2 | 80.4 |
| DYNAMIC | - | - | 87.6 | - | - | 79.4 | - | - | 73.6 | - | - | 48.2 | - |
| Our Method | 86.8 | 90.1 | 92.1 | 97.1 | 95.7 | 96.9 | 70.2 | 70.6 | 71.4 | 67.8 | 68.5 | 68.3 | 81.4 |

**Vision Evaluations.** To assess the generalization capability of our method on vision tasks, we report results under the 4Top1, 4Top2, and 6Top2 configurations in Tab. 3. Our method enhances the performance of GMoE across benchmarks, yielding an average performance increase of 1.0%. Moreover, the results show that our method surpasses the recent DYNAMIC MoE Guo et al. (2024).

## 4.3 ABLATION STUDIES

Tab. 4 ∼ Tab. 5 show ablation studies on MoE-LLaVA and Molmo. The results confirm the importance of modality-specific distribution-aware router (MsDaR) and vision-specific dynamic expert activation (VsDEA) modules. In contrast, removing all load balancing (ALB) yields negligible benefits.

The running time on MoE-LLaVA is shown in Tab. 6. While our method activates more experts for vision tail tokens, its inference time does not increase significantly. This stems from the all-to-all communication waiting principle: our method enhances expert activation without overburdening the slowest expert, thereby preserving inference time. Details of expert load is provided in B.1.

Table 4: Ablations on MoE-LLaVA-4Top2 with StableLM-1.6B.

| Method | GQA | SQA$^I$ | VQA$^T$ | POPE | MME | MMB | MM-Vet | Avg |
|---|---|---|---|---|---|---|---|---|
| MoE-LLaVA-4Top2 | 60.3* | 62.6 | 50.1 | 85.7 | 1318.2 | 60.2 | 26.9 | 57.6 |
| + MsDaR | 61.1* | 62.3 | 51.2 | 86.6 | 1324.3 | 59.9 | 27.9 | 58.2 |
| + MsDaR&VsDEA (LTDR) | 61.1* | 63.4 | 51.1 | 86.6 | 1363.5 | 60.6 | 29.9 | 58.8 |
| - ALB | 61.1* | 61.9 | 51.3 | 86.2 | 1324.2 | 60.4 | 28.5 | 58.2 |

Table 5: Ablations on Molmo-64Top8 with MolmoE-1B-7B.

| Method | ChartQA | DocVQA | AI2D | VQA | AndroidControl | CountBenchQA | Avg |
|---|---|---|---|---|---|---|---|
| MolmoE-1B-7B | 65.7 | 79.8 | 85.2 | 82.6 | 81.8 | 74.0 | 78.2 |
| +MsDaR | 67.8 | 80.7 | 86.4 | 83.0 | 81.7 | 76.9 | 79.4 |
| +MsDaR&VsDEA (LTDR) | 68.1 | 81.3 | 87.4 | 83.3 | 83.2 | 77.6 | 80.2 |
| - ALB | 65.9 | 80.2 | 85.3 | 82.9 | 81.5 | 74.7 | 78.4 |

Table 6: Running time on MoE-LLaVA-4Top2 with StableLM-1.6B.

| Method | GPU | GQA | SQA$^I$ | VQA$^T$ | POPE | MME | MMB | MM-Vet | Avg (s) |
|---|---|---|---|---|---|---|---|---|---|
| MoE-LLaVA-4Top2 | A800-80G | 1771 | 285 | 1265 | 1269 | 363 | 603 | 310 | 838 |
| Our method | A800-80G | 1698 | 301 | 1057 | 1116 | 317 | 595 | 310 | 770 |
| MoE-LLaVA-4Top2 | V100-30G | 2284 | 331 | 1277 | 1552 | 414 | 835 | 366 | 1008 |
| Our Method | V100-30G | 2252 | 368 | 1259 | 1530 | 401 | 825 | 363 | 1000 |

## 4.4 COMPARISON STUDIES

**Routing Strategies.** We compare with different routing strategies in Tab. 7. Please see B.2 for detailed implementation. Our method achieves state-of-the-art performance on GQA, VQA$^T$, POPE, MME and MM-Vet. On SQA$^I$, TASK and CLUSTER perform best, with our method close behind. Overall, our method attains the highest average score, demonstrating its superior performance.

**DeepSeekMoE.** We further compare with DeepSeekMoE Dai et al. (2024) in Tab. 8, which partitions $K$ experts into $mK$ finer experts and activates $mk$ of them. Experts are subdivided as $k_s$ shared experts for shared learning and $k_r$ routing experts for specialized learning. Evaluations under three distinct settings ($1_{1.0} + 3_{1.0}$, $1_{1.0} + 12_{0.25}$, $1_{1.0} + 16_{0.25}$) reveal that DeepSeekMoE underperforms our method, despite its increase specialization. This supports our hypothesis that vision and language TER differ, highlighting the need for tailored vision-language modality-aware routing strategies.

**Vision Token Selection in VsDEA.** We choose vision tail tokens (VTTs), whose RPV exceeds the mean RPV of all vision tokens. Two strategies are compared in Tab. 9: Vision head tokens (VHTs), whose RPV is below the mean RPV of all vision tokens. It selects a large proportion of tokens ($\approx$87%) compared to VTTs ($\approx$13%) . Instruction-aware Tokens (IATs). The attention scores between the instruction and vision tokens are used to identify the top 15% of vision tokens for VsDEA, ensuring the selected tokens are highly relevant to the given instructions. As shown in Tab. 9, enhancing expert activations for vision head tokens also improve the model's ability to learn visual information, although the benefits are less significant than those achieved with VTTs. Moreover, selecting a large proportion of tokens increases the training time cost. The influence of IATs is minimal, potentially due to noise from both visual and textual information, which may hinder its ability to reliably identify the most important vision tokens. Finally, we conduct quantitative comparisons against three fixed thresholds in B.5 to further demonstrate the consistent performance advantage of our method.

**Additional Comparison Studies.** We provide additional comparison studies in B, which includes: B.1 Expert Load: Explaining why our method introduces more experts but does not significantly increase inference time due to the all-to-all communication. B.2 Routing Strategies: Explaining the detailed implementation description of different routing strategies. B.3 Modality-aware MoE Architecture: Comparing our method with a modality-aware MoE architecture. B.4 Object Hallucination Evaluation: Assessing generation reliability across different model configurations. B.5 Vision Token Selection in VsDEA: Comparing our dynamic token selection with the fix-threshold token selection.

Table 7: **Comparison between different routing strategies on MoE-LLaVA-4Top2.** TASK, CLUSTER, INSTRUCT, DYNAMIC, STGC, MODALITY and DISTRIBUTION denote task routing Gururangan et al. (2021); Jain et al. (2024); Zhou et al. (2024), cluster routing Dou et al. (2023); Gou et al. (2023), instruction routing Chen et al. (2023a), dynamic routing Huang et al. (2024); Guo et al. (2024), conflicts mitigation routing Yang et al. (2024), modality-aware routing Chen et al. (2024a); Lin et al. (2024b); Nguyen et al. (2024), and distribution-aware routing Wang et al. (2020); Jin et al. (2023) respectively. All experiments are conducted on the MoE-LLaVA-4Top2 with StableLM-1.6B model. The best and second-best results are marked with **boldface** and underline.

| Method | GQA | SQA$^I$ | VQA$^T$ | POPE | MME | MMB | MM-Vet | Avg |
|---|---|---|---|---|---|---|---|---|
| MoE-LLaVA-4Top2 | 60.3$^*$ | 62.6 | 50.1 | 85.7 | 1318.2 | 60.2 | 26.9 | 57.6 |
| + TASK | 58.2$^*$ | **63.7** | 49.2 | 81.5 | 1306.3 | 59.5 | 25.2 | 56.2 |
| + CLUSTER | 57.0$^*$ | **63.7** | 50.3 | 86.1 | 1312.8 | 62.3 | 27.3 | 57.8 |
| + INSTRUCT | 58.1$^*$ | 63.2 | 50.0 | 85.9 | 1338.8 | 61.5 | 26.6 | 57.5 |
| + DYNAMIC | 61.0$^*$ | 62.1 | 49.2 | 85.7 | 1320.4 | **62.4** | 28.2 | 58.1 |
| + STGC | 60.9$^*$ | 62.6 | 50.7 | 85.9 | 1355.1 | 60.7 | 28.2 | 58.1 |
| + MODALITY | 60.4$^*$ | 61.6 | 49.2 | 85.9 | 1293.3 | 61.1 | 28.4 | 57.7 |
| + DISTRIBUTION | 60.1$^*$ | 62.6 | 50.5 | 86.5 | 1309.2 | 60.7 | 27.8 | 58.0 |
| + LTDR (Ours) | **61.1**$^*$ | 63.4 | **51.1** | **86.6** | **1363.5** | 60.6 | **29.9** | **58.8** |

Table 8: **Comparison with the "Shared Experts + Routed Experts" strategy of DeepSeekMoE** Dai et al. (2024) **on MoE-LLaVA-4Top2.** $S_s + R_r$ denotes S shared expert(s) of size s combined with R routing experts of size r. All experiments are conducted on the MoE-LLaVA-4Top2 with StableLM-1.6B model. The best and second-best results are marked with **boldface** and underline.

| Method | GQA | SQA$^I$ | VQA$^T$ | POPE | MME | MMB | MM-Vet | Avg |
|---|---|---|---|---|---|---|---|---|
| MoE-LLaVA-4Top2 ($4_{1.0}$) | 60.3$^*$ | 62.5 | 50.1 | 85.7 | 1318.2 | 60.2 | 26.9 | 57.6 |
| + $1_{1.0} + 3_{1.0}$ | **61.1**$^*$ | 62.5 | 50.4 | 86.2 | 1330.1 | 57.6 | 27.9 | 57.6 |
| + $1_{1.0} + 12_{0.25}$ | 60.4$^*$ | 62.7 | 50.4 | 86.2 | 1327.1 | 57.6 | 27.9 | 57.5 |
| + $1_{1.0} + 16_{0.25}$ | 60.9$^*$ | 62.8 | 49.9 | 86.2 | 1318.5 | 58.7 | 28.1 | 57.7 |
| + LTDR ($4_{1.0}$) | **61.1**$^*$ | **63.4** | **51.1** | **86.6** | **1363.5** | **60.6** | **29.9** | **58.8** |

Table 9: **Different dynamic vision token selection strategies in enhancing expert activation.** All experiments are conducted on the MoE-LLaVA-4Top2 with StableLM-1.6B model.

| Method | GQA | SQA$^I$ | VQA$^T$ | POPE | MME | MMB | MM-Vet | Avg |
|---|---|---|---|---|---|---|---|---|
| MoE-LLaVA-4Top2 | 60.3$^*$ | 62.6 | 50.1 | 85.7 | 1318.2 | 60.2 | 26.9 | 57.6 |
| VHTs | 61.0$^*$ | 62.0 | 50.3 | 86.3 | 1310.9 | 60.3 | 27.8 | 58.0 |
| IATs | 60.8$^*$ | 61.0 | 50.8 | 87.0 | 1307.8 | 59.7 | 26.9 | 57.7 |
| VTTs (Ours) | 61.1$^*$ | 63.4 | 51.1 | 86.6 | 1363.5 | 60.6 | 29.9 | 58.8 |

D Visualization Examples: Analyses include expert load and token activation map and visualization cases. Offering supporting evidence for the design choices behind our method.

## 4.5 Confidence intervals of the models across different random seeds

We assess StableLM, Qwen, and Phi-2 using three seeds to evaluate the consistency of our method. As shown in Tab. 10, our approach consistently outperforms all baselines across these evaluations.

## 5 Conclusion

We reveal the distinct token-to-expert routing (TER) distributions in vision-language tasks: language TER follows a uniform distribution, while vision TER exhibits a long-tailed distribution. This challenges the traditional load balancing mechanism in MoE: experts should receive an equal count of tokens to avoid a small number of experts gaining a disproportionately large share of preferences by the router. To address this, we propose the **L**ong-**T**ailed **D**istribution-aware **R**outer (LTDR) for vision-language TER, addressing two key challenges: (1) Modality-specific distribution-aware routing. We retain the load balancing mechanism for language TER but abandon it for vision TER,

Table 10: Confidence intervals of the MoE-LLaVA-4Top2 with StableLM-1.6B mode across seeds.

| MoE-LLaVA | GQA | ScienceQA | TextVQA | POPE | MME | MMBench | MM-Vet | Avg |
|---|---|---|---|---|---|---|---|---|
| StableLM | 60.3±0.20 | 62.4±0.20 | 50.0±0.15 | 85.5±0.21 | 1295.1±26.93 | 60.0±0.15 | 26.9±0.40 | 57.5±0.12 |
| +LTDR | 61.1±0.15 | 63.4±0.15 | 51.0±0.10 | 86.6±0.15 | 1342.9±24.66 | 60.7±0.12 | 29.6±0.25 | 58.8±0.06 |
| Qwen | 61.4±0.15 | 62.9±0.15 | 48.0±0.10 | 87.1±0.10 | 1286.2±21.00 | 59.6±0.15 | 25.4±0.17 | 57.4±0.00 |
| +LTDR | 61.6±0.20 | 62.8±0.06 | 48.9±0.15 | 87.3±0.06 | 1308.7±22.90 | 60.3±0.21 | 25.7±0.15 | 57.8±0.10 |
| Phi-2 | 61.2±0.20 | 68.4±0.21 | 51.3±0.23 | 86.2±0.21 | 1391.9±29.79 | 65.3±0.15 | 34.1±0.15 | 61.1±0.10 |
| +LTDR | 62.2±0.15 | 68.5±0.20 | 52.1±0.10 | 86.7±0.00 | 1414.5±26.45 | 66.7±0.20 | 34.2±0.20 | 61.7±0.06 |

enabling important vision tail tokens to be routed to specialized experts. (2) Vision-specific expert activation. Recognizing the importance of vision tail tokens, we employ an oversampling-like strategy, which increases the number of activated experts to ensure their thorough processing. To verify the effectiveness of our LTDR, we conduct extensive experiments on both vision-language and vision benchmarks. Experimental results verify the effectiveness of our approach.

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

# APPENDIX

We provide comprehensive supplementary materials, including implementation details, additional experiments, visualizations, and analyses, as follows:

- A: Implementation details of experiments.
- B: Additional comparison studies.
- C: Confidence studies.
- D: Visualization examples.
- E: Limitations of this work.
- E: Experiments conducted in response to the reviewers' comments (reformatted later).

## A  IMPLEMENTATION DETAILS

### A.1  BENCHMARKS

Our evaluation employs the vision-language benchmarks established by MoE-LLaVA Lin et al. (2024a) and Molmo Deitke et al. (2024), alongside the vision benchmarks used in GMoE Li et al. (2022). A detailed listing of these benchmarks is provided in Tab. 11.

Table 11: Our benchmarks on both vision-language and vision.

| Benchmark | Description |
|---|---|
| **Vision-Language Benchmarks for *MoE-LLaVA*** | |
| **Benchmark** | **Description** |
| GQA Hudson & Manning (2019) | *For visual perception capabilities of models through open-ended short answers.* |
| ScienceQA Lu et al. (2022) | *For zero-shot generalization of models on scientific question answering.* |
| TextVQA Singh et al. (2019) | *For text-rich visual question answering tasks.* |
| POPE Li et al. (2023) | *For degree of hallucination in model responses on three sampled subsets of COCO Lin et al. (2014).* |
| MME Fu et al. (2023) | *For visual perception of models with yes/no questions.* |
| MMBench Liu et al. (2023b) | *For robustness of model answers with all-round shuffling on multiple choice answers.* |
| MM-Vet Yu et al. (2023) | *For model capabilities in engaging in visual conversations on a diverse range of tasks and evaluates the correctness and helpfulness of the responses using the GPT-4 evaluation framework.* |
| **Vision-Language Benchmarks for *Molmo*** | |
| **Benchmark** | **Description** |
| ChartQA Masry et al. (2022) | *For question answering on charts and graphs.* |
| DocVQA Mathew et al. (2021) | *For answering questions posed on document images.* |
| AI2D Kembhavi et al. (2016) | *For question answering on scientific diagrams designed for elementary school level.* |
| VQA v2.0 Goyal et al. (2017) | *For answering open-ended questions about natural images.* |
| AndroidControl Li et al. (2024) | *For evaluating multi-modal models' ability to control smartphone apps based on screen images and instructions.* |
| CountBenchQA Beyer et al. (2024) | *Specifically designed to evaluate the object counting ability of vision-language models.* |
| **Vision Benchmarks for *GMoE*** | |
| **Benchmark** | **Description** |
| PACS Li et al. (2017) | *Consists of intersecting classes from Caltech256, Sketchy401 Sangkloy et al. (2016), TU-Berlin Eitz et al. (2012), and Google Images, providing a diverse set of visual domains.* |
| VLCS Albuquerque et al. (2019) | *Combines examples from five overlapping classes across VOC2007 Everingham et al. (2010), LabelMe Russell et al. (2008), Caltech-101 Griffin et al. (2007), and SUN Choi et al. (2010), offering a broad evaluation of cross-domain generalization.* |
| Office-Home Venkateswara et al. (2017) | *Contains approximately 15,500 images organized into 65 categories across four domains, making it suitable for assessing domain adaptation algorithms.* |
| DomainNet Peng et al. (2019) | *For the need of multi-source unsupervised domain adaptation research, featuring six domains and around 0.6 million images distributed across 345 categories.* |

## A.2 INSTRUCTION TUNING DATASETS

For instruction tuning, we utilize the mixed dataset from LLaVA Liu et al. (2023a) for the MoE-LLaVA and the academic dataset for Molmo Deitke et al. (2024). The structure and composition of these datasets are detailed in Tab. 12.

Table 12: Instruction-following data mixture.

| Data | Size | Response formatting prompts |
|---|---|---|
| *MoE-LLaVA* | | |
| LLaVA | 158K | - |
| ShareGPT | 40K | |
| VQAv2 | 83K | |
| GQA | 72K | Answer the question using a single word or phrase. |
| OKVQA | 9K | |
| OCRVQA | 80K | |
| A-OKVQA | 66K | Answer with the option's letter from the given choices directly. |
| TextCaps | 22K | Provide a one-sentence caption for the provided image. |
| RefCOCO | 48K | Provide a short description for this region. |
| VG | 86K | Provide the bounding box coordinate of the region this sentence describes. |
| **Total** | **665K** | |
| *Molmo* | | |
| VQA v2.0 | 440K | Open-ended question answering about general images. |
| TextVQA | 35K | Answer questions by reading text in images. |
| OK-VQA | 9K | Answer questions requiring external commonsense knowledge. |
| ChartQA | 28K | Question answering about charts and graphs. |
| DocVQA | 39K | Question answering on document images. |
| InfographicVQA | 24K | Information-seeking visual question answering. |
| AI2D | 15K | Question answering about scientific diagrams. |
| A-OKVQA | 17K | Commonsense reasoning about images. |
| AndroidControl | 300K | Evaluating the ability to control apps. |
| ScienceQA | 6K | Multimodal science question answering. |
| TabWMP | 23K | Answer questions based on tables and charts. |
| ST-VQA | 25K | Situated reasoning in visual question answering. |
| TallyQA | 250K | Evaluating complex counting ability. |
| **Total** | **605.5K** | |

## A.3 TRAINING CONFIGURATIONS

Our training configurations are based on MoE-LLaVA Lin et al. (2024a), Molmo Deitke et al. (2024), and GMoE Li et al. (2022). The detailed hyper-parameters and implementation specifics are provided in Tab. 13.

Table 13: Training Hyper-parameters.

| Epoch | Learning Rate | Learning Rate Schedule | Weight Decay | Load Balancing Loss Coefficient |
|---|---|---|---|---|
| 1 | 2e-5 | Cosine | 0.0 | 0.01 |

| Text Max Length | Batch Size per GPU | Train Step | Precision | The Number of $a$ in VsDEA |
|---|---|---|---|---|
| 2048 | 16 | original(others)/2000(Molmo) | Fp16 | 4(others)/12(Molmo) |

# B ADDITIONAL COMPARISON STUDIES

## B.1 EXPERT LOAD

Although our method activates more experts for processing vision tokens, it does not incur a significant increase in inference time. We attribute this efficiency to the all-to-all communication waiting principle inherent in MoE, where the overall inference speed is determined by the slowest expert (which handles the most tokens). To validate this hypothesis, we compare the expert load of MoE-LLaVA and our method on MME. As shown in Tab. 14, our method enhances expert activation without substantially increasing the slowest expert load, thereby preserving inference efficiency.

Table 14: **Comparison of expert load between MoE-LLaVA and our method on MME**. All experiments are conducted on the MoE-LLaVA-4Top2 with StableLM-1.6B model.

| Layer | MoE-LLaVA | | | | Our Method | | | | Slowest Expert Ratio |
|---|---|---|---|---|---|---|---|---|---|
| | E1 | E2 | E3 | E4 | E1 | E2 | E3 | E4 | (ours/baseline, %) |
| L0 | 1,253,363 | 2,226,698 | 398,040 | 1,156,335 | 1,198,348 | 1,029,670 | 1,875,783 | 931,000 | 84% |
| L2 | 997,557 | 1,134,398 | 1,768,130 | 1,134,651 | 1,878,331 | 1,066,184 | 963,695 | 1,126,482 | 106% |
| L4 | 1,184,666 | 173,013 | 1,415,384 | 2,261,376 | 1,067,353 | 74,966 | 2,985,489 | 906,990 | 132% |
| L6 | 488,128 | 1,656,115 | 2,580,364 | 309,985 | 3,126,276 | 711,835 | 711,004 | 485,621 | 121% |
| L8 | 345,515 | 538,817 | 76,701 | 4,073,395 | 160,092 | 165,625 | 4,431,660 | 277,250 | 108% |
| L10 | 1,510,384 | 2,546,079 | 89,496 | 888,421 | 718,307 | 539,202 | 56,546 | 3,721,096 | 146% |
| L12 | 1,048,012 | 221,306 | 487,068 | 3,278,533 | 567,314 | 1,707,754 | 994,087 | 1,765,437 | 53% |
| L14 | 4,727,048 | 63,570 | 238,961 | 6,364 | 885,210 | 1,744,495 | 205,466 | 2,199,600 | 46% |
| L16 | 387,689 | 3,458,834 | 677,254 | 510,819 | 647,612 | 1,028,262 | 2,296,453 | 1,062,287 | 66% |
| L18 | 1,204,376 | 1,606,444 | 726,607 | 1,497,214 | 1,062,718 | 2,199,400 | 473,470 | 1,299,001 | 136% |
| L20 | 830,383 | 2,315,369 | 1,294,702 | 594,244 | 1,355,671 | 842,243 | 2,217,780 | 618,905 | 95% |
| L22 | 1,347,728 | 55,666 | 387,303 | 3,245,075 | 2,397,151 | 901,297 | 1,291,264 | 444,928 | 73% |

## B.2 ROUTING STRATEGIES

TASK, CLUSTER, INSTRUCT, DYNAMIC, STGC, MODALITY-AWARE and DISTRIBUTION-AWARE denote task routing Gururangan et al. (2021); Jain et al. (2024); Zhou et al. (2024), cluster routing Dou et al. (2023); Gou et al. (2023), instruction routing Chen et al. (2023a), dynamic routing Huang et al. (2024); Guo et al. (2024), conflicts mitigation routing Yang et al. (2024), modality-aware routing Chen et al. (2024a); Lin et al. (2024b); Nguyen et al. (2024), and distribution-aware routing Wang et al. (2020); Jin et al. (2023) respectively. The details are illustrated as follows:

1. *TASK*. Similar to MoLA Zhou et al. (2024), aims to enhance similar routing for data from the same task while ensure distinct routing for data from different tasks. We conduct experiments using the LLaVA-mix-665k dataset, which is significantly different from the data used in MoLA. Empirically, we categorize the data into four task types: (1) *Caption*, where the instruction might be "provide a one-sentence caption for the provided image."; (2) *VQA*, with instructions like "answer the question using a single word or phrase."; (3) *OCR*, which includes all data from OCRVQA; and (4) *Region-aware*, where the instruction could be "provide a short description for this region." Each task type is assigned an expert label: 0 for *Caption*, 1 for *VQA*, 2 for *OCR*, and 3 for *Region-aware*. The dataset distributions are: *Caption* accounts for 3.5%, *VQA* for 61.6%, *OCR* for 12.8%, and *Region-aware* for 22.1%.

2. *CLUSTER*. Following MoCLE Gou et al. (2023), we encode instructions from different datasets using the all-MiniLM-L6-v2 [1] and cluster their embeddings using the $k$-means clustering algorithm. After clustering, in line with MoCLE's approach, we initialize $K$ learnable embeddings, where each embedding corresponds to a cluster center. When a sample belongs to the $k$-th cluster center, the $k$-th learnable embedding is extracted and passed to the router to predict routing scores. We set $k$-th=128, consistent with MoCLE's practice, and do not incorporate the load balancing loss.

3. *INSTRUCT*. Following the approach of LoRA-MoE Chen et al. (2023a), we compute the average of instruction token representations for each instance and use this as input to predict its routing scores across experts. Based on these routing scores, the Top-$k$ experts are selected for each sample to generate the final prediction. In alignment with LoRA-MoE's methodology, we do not include the load balancing loss in our implementation.

---

[1] https://huggingface.co/sentence-transformers/all-MiniLM-L6-v2.

4. **DYNAMIC**. Drawing inspiration from DYNMOE Guo et al. (2024), we highlight its gating mechanism, which allows each token to dynamically determine the number of experts to activate. Additionally, an adaptive process automatically adjusts the number of experts during training. We reference its experimental results on the MoE-LLaVA-4Top2 with StableLM-1.6B model, where an average of 1.25 experts out of 4 are activated per token.

5. **STGC**. STGC Yang et al. (2024) employs token-level gradients to identify conflicting tokens within experts. Additionally, it introduces a regularization loss designed to encourage conflicting tokens to route away from their current experts to alternative ones, thereby minimizing interference among tokens within the same expert. We reference their experimental results on the MoE-LLaVA-4Top2 using the StableLM-1.6B model.

6. **MODALITY**. Following MoMa Lin et al. (2024b), we partition experts into two groups dedicated to vision and language, respectively. To maintain the total number of experts $K$ (set to 4 in this study) and the number of activated experts $k$ (set to 2 in this study), we designate 2 experts as vision experts and the remaining 2 as language experts, activating 1 expert from each modality group.

7. **DISTRIBUTION**. Following RIDE Wang et al. (2020), we cluster tokens into six categories via k-means (matching the 4Top2 expert paths, $C_4^2$. Each category is mapped to a specific expert pair (e.g., category $1 \to$ experts 1 & 2). Tokens of each category are directed to their assigned expert pair.

## B.3 MODALITY-AWARE MoE ARCHITECTURE

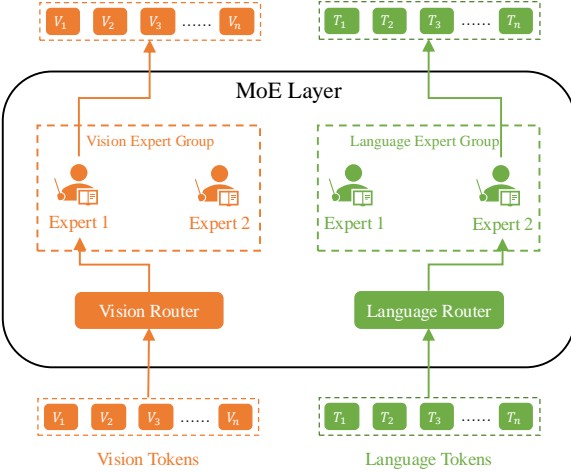

Figure 3: The modality-aware MoE architecture which divides experts for vision and language.

To compare with existing modality-aware MoE architectures as shown in Fig. 3, such as Chen et al. (2024a); Lin et al. (2024b); Nguyen et al. (2024), we adopt the following configurations:

1. **MoE-LLaVA-v2Top1-t2Top1.** We partition experts into two groups dedicated to vision and language, respectively. As illustrated in Fig. 3, to maintain the total number of experts $K$ (set to 4 in this study) and the number of activated experts $k$ (set to 2 in this study), we designate 2 experts as vision experts and the remaining 2 as language experts, activating 1 expert from each modality group.

2. **MoE-LLaVA-v4Top2-t4Top2.** We first expand the 4 experts into 8 by splitting FFN intermediate hidden dimension Dai et al. (2024), assigning 4 as vision experts and the remaining 4 as language experts, and activating 2 experts from each modality group.

3. **MoE-LLaVA-v2Top1-t2Top1-MsDaR.** This configuration is similar to *MoE-LLaVA-v2Top1-t2Top1*, but it removes the expert load balancing constraint for the vision expert group (similar to the distribution-aware router in our proposed approach, LTDR).

4. **MoE-LLaVA-v2Top1-t2Top1-MsDaR-shared.** This setup is akin to *MoE-LLaVA-v2Top1-t2Top1-MsDaR*, but includes a shared expert for world knowledge Dai et al. (2024).

The results in Tab. 15 indicate that the modality-aware MoE does not enhance the performance of MoE-LLaVA. Moreover, increasing the number of experts results in a significant performance decline, suggesting that a larger number of experts exacerbates expert load balancing issues, which negatively impacts vision TER. When the expert load balancing constraint is removed for the vision expert group, *MoE-LLaVA-v2Top1-t2Top1-MsDaR* shows improved performance compared to MoE-LLaVA-4Top2, validating the effectiveness of our MsDaR module. Finally, the addition of an extra expert in *MoE-LLaVA-v2Top1-t2Top1-MsDaR-shared* does not yield benefits, highlighting the distinctions between vision-language MoE and language MoE Dai et al. (2024).

Table 15: **Comparison with modality-aware MoE** Chen et al. (2024a); Lin et al. (2024b); Nguyen et al. (2024). All experiments are conducted on the MoE-LLaVA-4Top2 with StableLM-1.6B model.

| Method | GQA | SQA$^I$ | VQA$^T$ | POPE | MME | MMB | MM-Vet | Avg |
|---|---|---|---|---|---|---|---|---|
| MoE-LLaVA-4Top2 | 60.3$^*$ | 62.6 | 50.1 | 85.7 | 1318.2 | 60.2 | 26.9 | 57.6 |
| *Modality-aware MoE* | | | | | | | | |
| MoE-LLaVA-v2Top1-t2Top1 | 60.4$^*$ | 61.6 | 49.2 | 85.9 | 1293.3 | 61.1 | 28.4 | 57.7 |
| MoE-LLaVA-v4Top2-t4Top2 | 60.3$^*$ | 58.6 | 46.8 | 85.7 | 1296.6 | 55.4 | 26.4 | 55.5 |
| MoE-LLaVA-v2Top1-t2Top1-MsDaR | 60.9$^*$ | 61.3 | 51.0 | 86.5 | 1324.5 | 61.1 | 28.4 | 58.2 |
| MoE-LLaVA-v2Top1-t2Top1-MsDaR-shared | 61.0$^*$ | 62.5 | 51.3 | 86.5 | 1333.8 | 60.0 | 28.0 | 58.2 |
| Our Method | 61.1$^*$ | 63.4 | 51.1 | 86.6 | 1363.5 | 60.6 | 29.9 | 58.8 |

## B.4 Object Hallucination Evaluation

We employ the evaluation pipeline of POPE Li et al. (2023), a polling-based query methodology, to assess object hallucination in MoE-LLaVA. As shown in Tab. 16, our proposed method demonstrates superior performance compared to baseline model, suggesting its enhanced capability in generating object descriptions that are more consistent with the visual content of the given images.

Table 16: **Zero-shot object hallucination evaluation results.** "L", "V" and "S" respectively represent LLaMA, Vicuna and StableLM. All "Sparse Model" methods use the configure 4Top2.

| Method | LLM | Adersarial | | | Popular | | | Random | | |
|---|---|---|---|---|---|---|---|---|---|---|
| | | Acc | F1-Score | Yes | Acc | F1-Score | Yes | Acc | F1-Score | Yes |
| *Dense Model* | | | | | | | | | | |
| mPLUG-Owl Ye et al. (2023) | L-7B | 82.4 | 81.6 | 45.2 | 85.5 | 84.3 | 42.1 | 86.3 | 85.3 | 42.3 |
| MM-GPT Gong et al. (2023) | L-7B | 50.0 | 66.7 | 100.0 | 50.0 | 66.7 | 100.0 | 50.0 | 66.7 | 100.0 |
| LLaVA-1.5 Liu et al. (2023a) | V-13B | 85.5 | 84.4 | 43.3 | 87.4 | 86.2 | 41.3 | 88.0 | 87.1 | 41.7 |
| *Sparse Model* | | | | | | | | | | |
| MoE-LLaVA-4Top2 Lin et al. (2024a) | S-1.6B | 86.9 | 85.7 | 41.7 | 85.3 | 84.2 | 43.5 | 88.0 | 87.1 | 41.6 |
| Our Method | S-1.6B | 85.8 | 84.2 | 43.3 | 86.5 | 86.0 | 41.4 | 88.0 | 87.2 | 41.3 |

## B.5 Vision Token Selection in VsDEA.

Our dynamic threshold for identifying vision tail tokens (VTTs) is computed per image as the mean RPV of its tokens, enabling automatic adaptation to the RPV distribution variations. This approach consistently identifies approximately 13% of vision tokens (576 tokens per image) as VTTs. As illustrated in Fig. 7, datasets like GQA, MMBench, and TextVQA exhibit distinct RPV distributions with unique means, highlighting the limitation of a fixed threshold. Quantitative comparisons in Tab. 17 confirm the consistent advantage of our dynamic method over fixed thresholds on the MoE-LLaVA with StableLM-1.6B model, which show scenario-dependent performance fluctuations.

Table 17: Comparison of fixed thresholds vision tail token selection in VsDEA.

| Ratio(%) | GQA | ScienceQA | TextVQA | POPE | MME | MMBench | MM-Vet | Avg |
|---|---|---|---|---|---|---|---|---|
| 10% | 61.0 | 62.2 | 50.9 | 87.2 | 1321.5 | 60.4 | 25.5 | 57.9 |
| 15% | 61.2 | 62.2 | 50.8 | 86.7 | 1332.1 | 61.4 | 25.3 | 57.9 |
| 20% | 61.1 | 62.8 | 50.1 | 86.1 | 1313.1 | 58.9 | 25.1 | 57.4 |
| Our Method≈13% | 61.1 | 63.4 | 51.1 | 86.6 | 1363.5 | 60.6 | 29.9 | 58.8 |

## C CONFIDENCE STUDIES

### C.1 QUANTIFIED VSDEA COMPUTATION AND MEMORY OVERHEAD

We quantify VsDEA's compute and memory usage using a V100-30G GPU. The results are shown in Tab. 18. It is worth noting that our method and the baseline show almost identical metrics.

Table 18: VsDEA computation and memory on the MoE-LLaVA-4Top2 with StableLM-1.6B model.

| Model | GQA | ScienceQA | TextVQA | POPE | MME | MMBench | MM-Vet | Avg |
|---|---|---|---|---|---|---|---|---|
| | | | Wall-clock Time per Step (s) | | | | | |
| MoE-LLaVA-4Top2 | 0.18 | 0.16 | 0.26 | 0.17 | 0.17 | 0.19 | 1.68 | 0.40 |
| our Method | 0.18 | 0.17 | 0.25 | 0.17 | 0.17 | 0.19 | 1.67 | 0.35 |
| | | | Memory Used (G) & GPU Util (%) | | | | | |
| MoE-LLaVA-4Top2 | 9.30 & 65 | 9.08 & 61 | 10.43 & 63 | 8.63 & 67 | 9.11 & 66 | 10.47 & 64 | 9.08 & 31 | - |
| Our Method | 9.32 & 67 | 9.03 & 57 | 10.43 & 59 | 8.63 & 67 | 9.11 & 67 | 10.47 & 65 | 9.08 & 33 | - |

### C.2 ABLATION STUDIES ON MODELS OF VARYING SCALES.

We perform studies with Qwen and Phi-2. The Tab. 19 aligns with the findings with StableLM-1.6B.

Table 19: Ablation studies on models of varying scales.

| MoE-LLaVA | GQA | ScienceQA | TextVQA | POPE | MME | MMBench | MM-Vet | Avg |
|---|---|---|---|---|---|---|---|---|
| | | | | Qwen-1.8B | | | | |
| Qwen-1.8B | 61.5 | 63.1 | 48.0 | 87.0 | 1291.6 | 59.7 | 25.3 | 57.4 |
| +MsDaR | 61.6 | 62.9 | 48.6 | 87.1 | 1315.6 | 60.0 | 25.6 | 57.6 |
| +MsDaR&VsDEA | 61.6 | 62.8 | 48.9 | 87.2 | 1334.2 | 60.5 | 25.5 | 57.8 |
| | | | | Phi2-2.7B | | | | |
| Phi2-2.7B | 61.4 | 68.5 | 51.4 | 86.3 | 1423.0 | 65.2 | 34.3 | 61.2 |
| +MsDaR | 61.8 | 68.6 | 51.8 | 86.7 | 1425.3 | 66.2 | 34.5 | 61.6 |
| +MsDaR&VsDEA | 62.2 | 68.5 | 52.0 | 86.7 | 1440.8 | 66.7 | 34.0 | 61.7 |

## D VISUALIZATION EXAMPLES

### D.1 EXPERT LOAD AND TOKEN ACTIVATION MAP

We analyze the distribution of expert load. MoE-LLaVA with StableLM-1.6B serves as the baseline model, and MoE-LLaVA+LTDR denotes our method. The expert load of total tokens and image-text tokens is shown in Fig. 4 ∼Fig. 5, respectively. Fig. 4 indicates that our method does not significantly amplify the expert load imbalance compared to the baseline, which aligns with the analyses in B.1. Fig. 5 depicts that language tokens follow a relatively uniform distribution. In contrast, our method eliminates the vision load balancing; the resulting imbalanced distribution of vision tokens shows that more vision tokens select specialized experts. This supports our hypothesis that the TER for language tokens adheres to a uniform distribution, whereas the TER for vision tokens follows a long-tailed distribution. Our method dynamically adjusts token-to-expert paths to improve the language expert load balancing and vision expert specialization, better handling modality-specific distributions. Fig. 6 shows token activation maps. Our method significantly changes the original token activation path.

### D.2 ROUTING PROBABILITY VARIANCE

We also compare the token routing probability variance (RPV) between vision head tokens and vision tail tokens across GQA Hudson & Manning (2019), MMBench Liu et al. (2023b) and TextVQA Singh et al. (2019). As illustrated in Fig. 7, for images that yield 576 tokens from CLIP, $\text{Mean}(\text{RPV}(\mathcal{V}))$ denotes the mean RPV of all vision tokens $\mathcal{V}$, $\mathcal{V}_{head}$ and $\mathcal{V}_{tail}$ denote vision head tokens and vision tail tokens. The bars in the figure is the mean token count with RPV ranging from left to right for images (*e.g.*, in the upper left figure, the count of tokens with RPV ranging from 0.00 to 0.01 is 442).

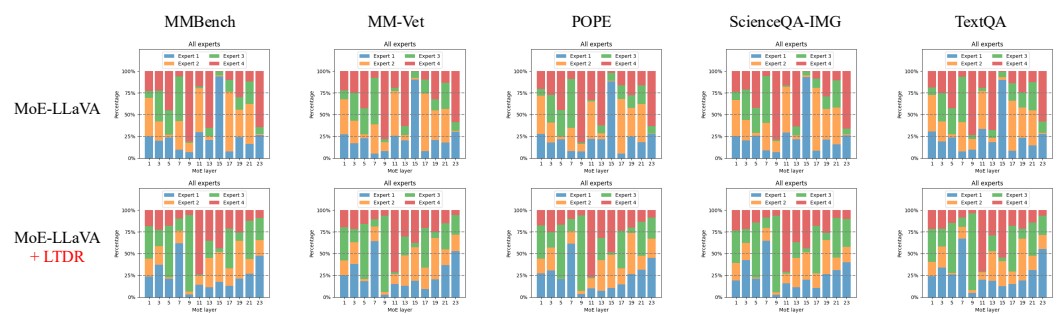

Figure 4: Expert load of total tokens.

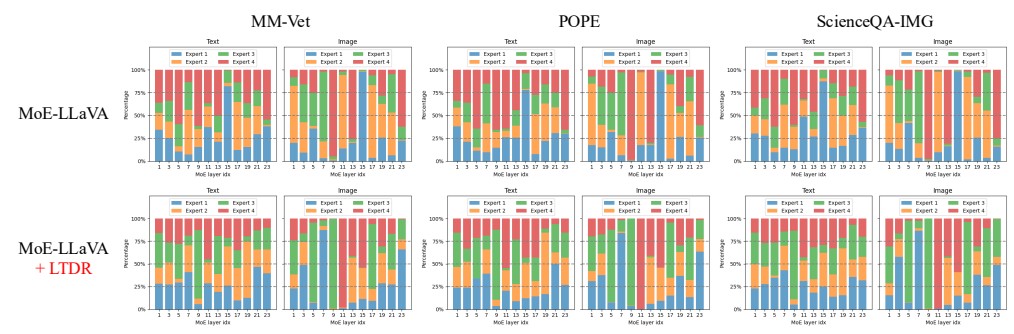

Figure 5: Expert load of image and text tokens.

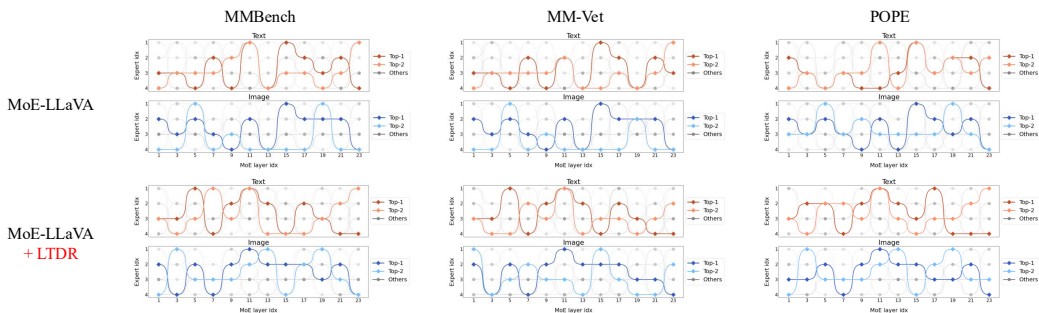

Figure 6: Token activation maps.

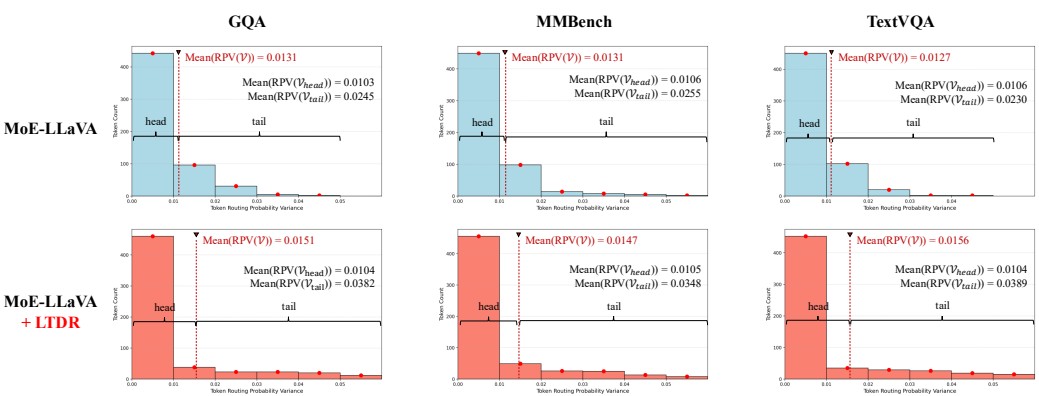

Figure 7: The Routing probability variance distribution of Vision head tokens and vision tail tokens.

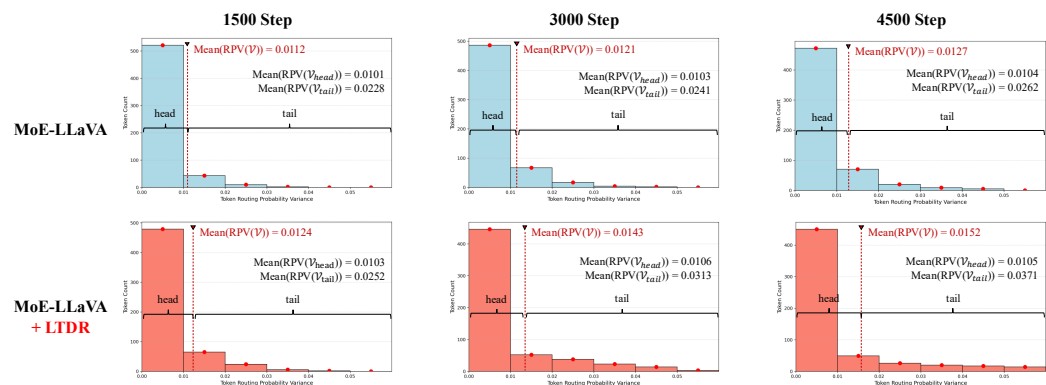

Figure 8: The Routing probability variance distribution at three training steps on TextVQA.

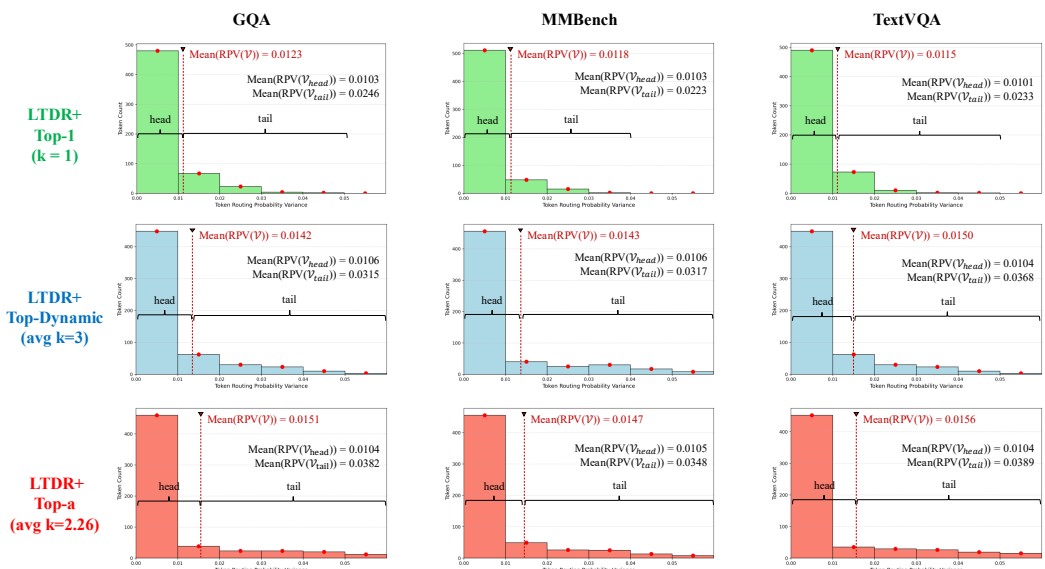

Figure 9: The Routing probability variance distribution cross routers.

Our method significantly increases the mean RPV of vision tail tokens. Given that RPV reflects the TER probability distribution, these results demonstrate that vision tail tokens gain the ability to select their specialized experts. Meanwhile, the mean RPV of vision head tokens remains nearly unchanged, indicating that vision head tokens are not affected.

Moreover, we add three sets of RPV distribution visualization comparisons: training stages RPV (Fig. 8), cross-router RPV (Fig. 9), and cross-backbone RPV (Fig. 10). The RPV distributions at 1,500, 3,000, and 4,500 training steps on TextVQA indicate that, although both the baseline and our method improve mean RPV, our approach achieves substantially greater enhancement. This suggests that the baseline has limited capacity to effectively capture critical visual information. Cross-Router: In the Top-Dynamic setting, expert activation is determined by token score ranking and expert capacity, where expert capacity is defined as $num_{\text{token}}/num_{\text{experts}} \times 3$. The results indicate that the mean RPV of the vision tail under Top-1 is significantly lower than in other configurations, suggesting that increasing the number of parameters can enhance vision tail token learning. However, despite Top-Dynamic having the largest number of parameters, it does not outperform Top-a in terms of vision tail RPV, implying that simply increasing parameter count does not guarantee performance improvement. Cross-Backbone: The choice of backbone has a notable impact on mean RPV. Among the evaluated backbones, Phi-2 achieves the highest mean RPV, followed by StableLM, with Qwen yielding the lowest. The performance of the models aligns with this ranking.

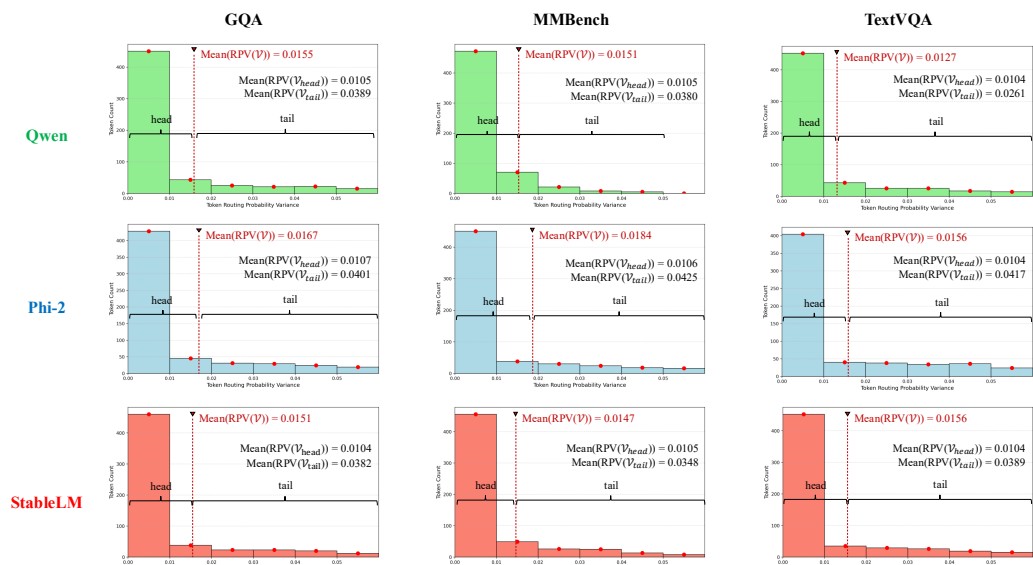

Figure 10: The Routing probability variance distribution cross backbones.

### D.3 VISUALIZATION CASES.

Our visualizations in Fig. 11 reveal that vision tail tokens focus on critical instruction-aware image patches, capturing question-relevant visual information which enhances answer accuracy.

## E LIMITATIONS

As the vision tail tokens are only identified in one batch of vision tokens in each iteration, it will inevitably misjudge some unimportant vision tokens as vision tail tokens, affecting the model's reception of key information. Therefore, although it has been demonstrated that the proposed solution can solve the expert specialization problem of vision tail tokens by strengthening vision tail token routing, the solution still has room for improvement because the identification of vision tail tokens is challenging. In the future, it may be an interesting direction to explore whether the vision tail tokens and their optimal experts can be determined from an optimization perspective.

## F EXPERIMENTS CONDUCTED IN RESPONSE TO THE REVIEWERS' COMMENTS

### F.1 MOLMO INFERENCE-TIME LATENCY

We calculated the inference latency of Molmo. As shown in Tab. 20, there is no significant difference in inference latency between the configurations with $a = 12$ (LTDR) and $k = 8$ (Molmo).

Table 20: Molmo inference-time latency on the A800-80G GPU.

| Method | GPU | ChartQA | DocVQA | AI2D | VQA | AndroidControl | CountBenchQA | Avg (s) |
|---|---|---|---|---|---|---|---|---|
| Molmo | A800-80G | 184 | 200 | 190 | 196 | 89 | 705 | 261 |
| +LTDR | A800-80G | 187 | 203 | 194 | 200 | 90 | 716 | 265 |

## F.2 STRATEGY-SWAP ABLATION ON VISION AND LANGUAGE

We perform a strategy-swap ablation on the language side while keeping the vision-side strategy fixed. As summarized in Tab. 21, removing the load balancing loss on the language side (language+MsDaR) results in fluctuating performance, yielding only a 0.2% average improvement over the vanilla model. Moreover, introducing text-specific dynamic expert activation (TsDEA) does not lead to consistent performance gains. In contrast, removing the load balancing loss on the vision side (vision+MsDaR) yields stable improvements, and further incorporating vision-specific dynamic expert activation (VsDEA) provides additional gains, achieving an average improvement of 1.2%.

Table 21: Strategy-swap ablation on vision and language.

| MoE-LLaVA_StableLM | GQA | ScienceQA | TextVQA | POPE | MME | MMBench | MM-Vet | Avg |
|---|---|---|---|---|---|---|---|---|
| Vanilla | 60.3 | 62.6 | 50.1 | 85.7 | 1318.2 | 60.2 | 26.9 | 57.6 |
| **Language** | | | | | | | | |
| + MsDaR | 60.8 | 62.0 | 50.2 | 85.9 | 1254.1 | 60.0 | 28.0 | 57.8 |
| + MsDaR&TsDEA | 60.1 | 61.2 | 50.4 | 86.2 | 1282.4 | 60.1 | 28.6 | 57.8 |
| **Vision** | | | | | | | | |
| + MsDaR | 61.1 | 62.3 | 51.2 | 86.6 | 1324.3 | 59.9 | 27.9 | 58.2 |
| + MsDaR&VsDEA | 61.1 | 63.4 | 51.1 | 86.6 | 1363.5 | 60.6 | 29.9 | 58.8 |

## F.3 BROADER CROSS-ROUTER PERFORMANCE

In addition to comparing the performance of different backbones in Section. 4.2, we also report the cross-router results, as summarized in Tab. 22. In the Top-Dynamic setting, expert activation is determined by token-score ranking and expert capacity, where the expert capacity is defined as $num_{\text{token}}/num_{\text{experts}} \times 3$. LTDR with Top-1 achieves a 0.9% improvement over the vanilla model. Although dynamic routers provide additional gains, their performance remains lower than that of LTDR combined with Top-a.

Table 22: Broader cross-router performance.

| MoE-LLaVA_StableLM | GQA | ScienceQA | TextVQA | POPE | MME | MMBench | MM-Vet | Avg |
|---|---|---|---|---|---|---|---|---|
| Vanilla+Top-1 (k=1) | 58.6 | 55.8 | 45.0 | 85.2 | 1245.3 | 56.2 | 27.2 | 54.7 |
| Vanilla+Top-2 (k=2) | 60.3 | 62.6 | 50.1 | 85.7 | 1318.2 | 60.2 | 26.9 | 57.6 |
| LTDR+Top-1 (k=1) | 59.7 | 58.2 | 45.6 | 85.8 | 1302.9 | 57.1 | 27.0 | 55.6 |
| LTDR+Dynamic (avg k=3) | 60.8 | 62.1 | 51.1 | 86.8 | 1332.8 | 60.2 | 28.9 | 58.3 |
| LTDR+Top-a (avg k=2.26) | 61.1 | 63.4 | 51.1 | 86.6 | 1363.5 | 60.6 | 29.9 | 58.8 |

## F.4 COMPARISON WITH SIMPLER SHARPENING VISION TOKEN ROUTING METHODS

To evaluate the sharpening of vision token routing, we compare it with several simpler routing strategies, including temperature scaling, entropy regularization, and heuristic sharpening. The results are summarized in Tab. 23. Temperature-based: Gumbel-Softmax with a temperature of 0.7. Entropy-based: standard entropy regularization to encourage more confident routing. Heuristic-based: variance constraints on the distribution of vision token routing. Simply sharpening the vision token routing does not yield significant improvements. We hypothesize that this is due to the distinct routing behaviors of head and tail vision tokens: tail tokens are relatively few, so sharpening their routing helps direct them to appropriate experts more effectively; in contrast, head tokens are abundant and already learn sufficiently from multiple experts, so further sharpening may actually impede the learning of tail tokens.

Table 23: Comparison with simpler sharpening vision token routing methods.

| MoE-LLaVA_StableLM | GQA | ScienceQA | TextVQA | POPE | MME | MMBench | MM-Vet | Avg |
|---|---|---|---|---|---|---|---|---|
| Vanilla | 60.3 | 62.6 | 50.1 | 85.7 | 1318.2 | 60.2 | 26.9 | 57.6 |
| Temperature-based | 60.9 | 62.8 | 50.2 | 86.0 | 1302.5 | 60.1 | 26.8 | 57.8 |
| Entropy-based | 61.1 | 62.4 | 50.8 | 86.2 | 1330.7 | 60.0 | 27.2 | 57.9 |
| Heuristic-based | 61.1 | 62.5 | 50.3 | 85.8 | 1299.6 | 60.2 | 26.6 | 57.7 |
| LTDR | 61.1 | 63.4 | 51.1 | 86.6 | 1363.5 | 60.6 | 29.9 | 58.8 |

## F.5 GENERALIZABILITY ACROSS DATASETS/MODELS

We compare the mean RPV using thresholds of 10%, 15%, and 20%, as the observed mean RPV (13%) falls within this range. This allows us to visually assess how small variations in the threshold affect performance. We evaluate these fixed-proportion thresholds on the MoE-LLaVA with StableLM-1.6B model across multiple test datasets, as detailed in Appendix B.5. Here, we further present generalizability experiments on the MoE-LLaVA with Phi-2-2.7B model, as shown in Tab. 24. The results show that mean-RPV consistently outperforms the comparison thresholds. We also compare mean-RPV with the learnable strategy in Tab. 25; however, the tuning method yields performance that lies between that of the original model and the LTDR-based model.

Table 24: Generalizability across different models.

| MoE-LLaVA_StableLM | GQA | ScienceQA | TextVQA | POPE | MME | MMBench | MM-Vet | Avg |
|---|---|---|---|---|---|---|---|---|
| Phi-2 with 10% | 62.1 | 68.3 | 51.6 | 86.5 | 1406.3 | 65.6 | 33.7 | 61.3 |
| Phi-2 with 15% | 62.0 | 68.4 | 51.8 | 86.6 | 1423.1 | 66.3 | 34.0 | 61.5 |
| Phi-2 with 20% | 61.3 | 67.8 | 51.3 | 86.2 | 1414.5 | 66.5 | 34.1 | 61.2 |
| Phi-2 with mean-RPV | 62.2 | 68.5 | 52.0 | 86.7 | 1440.8 | 66.7 | 34.0 | 61.7 |

Table 25: Comparison with systematic tuning policy.

| MoE-LLaVA_StableLM | GQA | ScienceQA | TextVQA | POPE | MME | MMBench | MM-Vet | Avg |
|---|---|---|---|---|---|---|---|---|
| Vanilla | 60.3 | 62.6 | 50.1 | 85.7 | 1318.2 | 60.2 | 26.9 | 57.6 |
| Tuning-based | 61.0 | 62.1 | 50.6 | 86.3 | 1322.3 | 59.9 | 28.9 | 58.1 |
| LTDR | 61.1 | 63.4 | 51.1 | 86.6 | 1363.5 | 60.6 | 29.9 | 58.8 |

## F.6 IMPACT OF REDUCING LOAD BALANCING

We analyze the effect of reducing load balancing, with the results presented in Tab. 26. Specifically, we decrease the vision-side load balancing coefficient from 0.01 to 0.001. The results indicate that this adjustment is less effective than removing load balancing entirely.

## F.7 THE LINK BETWEEN RPV AND TOKEN INFORMATIVENESS

We evaluate the link between RPV and token informativeness from two perspectives: 1) Performance-based analysis (interpretability). Higher model performance suggests that the corresponding tokens carry richer information. As described in Section 4.4, we sort tokens by RPV in descending order and classify those above the mean RPV as vision tail tokens (13%) and the remaining ones as vision head tokens (87%). Although vision head tokens are far more numerous than tail tokens, applying the VsDEA strategy to head tokens yields substantially lower performance than applying it to tail tokens. This performance gap indicates that high-RPV vision tail tokens encode more informative content. 2) Statistics-based analysis (statistical). A token group with a higher mean L2 norm of its vector representations reflects richer underlying information. We compare the mean L2 norm of the top 13%

Table 26: Impact of reducing load balancing.

| MoE-LLaVA_StableLM | GQA | ScienceQA | TextVQA | POPE | MME | MMBench | MM-Vet | Avg |
|---|---|---|---|---|---|---|---|---|
| Vanilla | 60.3 | 62.6 | 50.1 | 85.7 | 1318.2 | 60.2 | 26.9 | 57.6 |
| Reducing-based | 60.2 | 62.4 | 50.9 | 86.3 | 1315.7 | 59.1 | 28.4 | 57.8 |
| LTDR | 61.1 | 63.4 | 51.1 | 86.6 | 1363.5 | 60.6 | 29.9 | 58.8 |

vision token vectors (vision tail tokens) with those of tokens ranked between the top 13%–26% and 26%–39% by RPV. As shown in Tab. 27, the first group exhibits a higher mean L2 norm than the latter two, providing additional evidence that high-RPV vision tail tokens carry richer information.

Table 27: Statistics-based analysis.

| MoE-LLaVA_StableLM | mean L2 norm |
|---|---|
| Top-13% | 0.3158 |
| Top-13% - Top-26% | 0.2124 |
| Top-26% - Top-39% | 0.1475 |

### F.8 Average Top-k, Activated Parameters and Performance during Inference

The detailed configurations of expert activation and parameter counts are provided in Tab. 28.

Table 28: Average Top-k, activated parameters and performance during inference.

| MoE-LLaVA_StableLM | Top-k | Activated Parameters (Billions) | Average Performance |
|---|---|---|---|
| Vanilla | 2 | 4.9 | 57.6 |
| LTDR | 2.26 | 5.2 | 58.8 |

### F.9 Scalability to higher vision-token scenarios

We evaluate inference performance on a multi-image visual question answering task, comparing our method with the baseline in terms of both accuracy and inference speed. Specifically, we use the NLVR2 dataset, in which each sample consists of two images and a textual statement, and the model must determine whether the statement correctly describes both images. Experimental results, presented in Tab. 29, show that our method outperforms the baseline by 1.5% in scenarios involving a higher number of vision tokens, while maintaining comparable inference time and introducing no significant latency overhead.

### F.10 Experimental Scale and Generalization

We conduct experiments using an expanded training dataset to provide a more comprehensive comparison. Specifically, we incorporate the Open-LLaVA-NeXT training set, which adds 350K samples to the original 665K samples. Results, summarized in Tab. 30, demonstrate that our method remains effective at larger scales and outperforms the baseline on the full 1,021K-sample dataset.

## G The Usage of Large Language Models

We only use large language models to assist and polish paper writing.

Table 29: Scalability on NLVR2.

| MoE-LLaVA_StableLM | NLVR2 | Inference Time (s) |
|---|---|---|
| Vanilla | 52.9 | 1121 |
| LTDR | 54.4 | 1124 |

Table 30: Validation on large-scale instruction-tuning datasets.

| MoE-LLaVA_StableLM | Data | GQA | ScienceQA | TextVQA | POPE | MME | MMBench | MM-Vet | Avg |
|---|---|---|---|---|---|---|---|---|---|
| Vanilla | 665K | 60.3 | 62.6 | 50.1 | 85.7 | 1318.2 | 60.2 | 26.9 | 57.6 |
| LTDR | 665K | 61.1 | 63.4 | 51.1 | 86.6 | 1363.5 | 60.6 | 29.9 | 58.8 |
| Vanilla | 1021K | 61.0 | 63.0 | 51.2 | 86.7 | 1360.3 | 61.0 | 29.2 | 58.6 |
| LTDR | 1021K | 61.4 | 65.4 | 52.5 | 87.3 | 1409.2 | 62.4 | 34.8 | 60.6 |

# Image-text Visualization Cases and Model Outputs

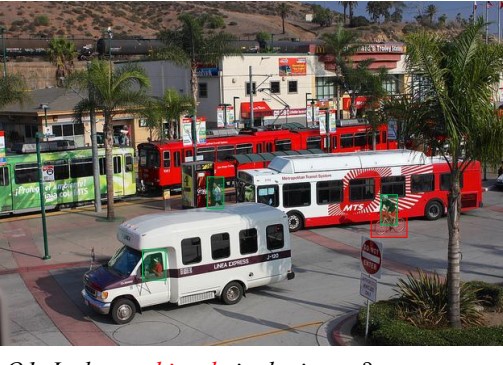 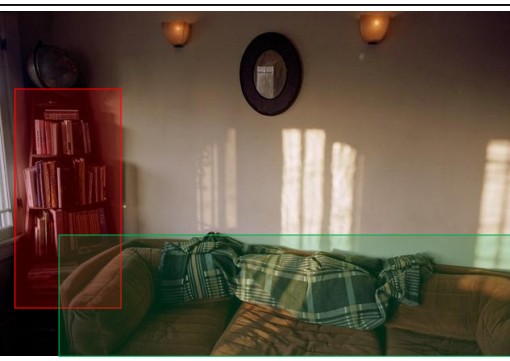

*Q1: Is there a bicycle in the image?*
MoE-LLaVA: No
MoE-LLaVA+LTDR: Yes

*Q2: Is there a person in the image?*
MoE-LLaVA: No
MoE-LLaVA+LTDR: Yes

*Q1:Which kind of furniture is wooden?*
MoE-LLaVA: Couch
MoE-LLaVA+LTDR: Bookcase

*Q2:What is the brown item of furniture?*
MoE-LLaVA: Mirror
MoE-LLaVA+LTDR: Couch

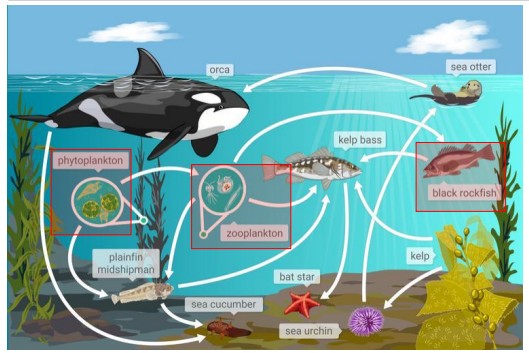 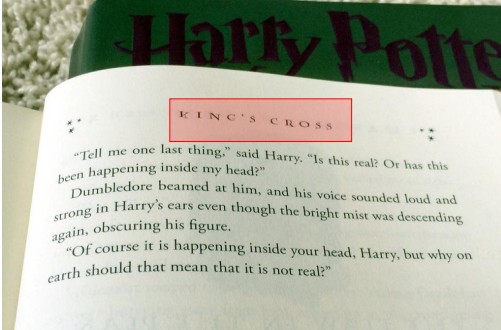

*Q1: The image shows a python code. Is the output of the code 'x is larger than 10'?*
MoE-LLaVA: Yes
MoE-LLaVA+LTDR: No

*Q2: The image shows a python code. Is the output of the code 'x is smaller than 10'?*
MoE-LLaVA: No
MoE-LLaVA+LTDR: Yes

*Q1: Which country has a below-average profit margin?*
MoE-LLaVA: South Korea
MoE-LLaVA+LTDR: India

*Q2: Which country has the highest profit margin?*
MoE-LLaVA: USA
MoE-LLaVA+LTDR: Australia

*Q: Which of these organisms contains matter that was once part of the phytoplankton?*
MoE-LLaVA: Sea Otter
MoE-LLaVA+LTDR: Black Rockfish

*Q1: What is the name of this chapter?*
MoE-LLaVA: Harry Potte
MoE-LLaVA+LTDR: King's Cross

Figure 11: Visualization Cases on benchmarks of MoE-LLaVA.