# OpenReview forum: "Long-Tailed Distribution-Aware Router For Mixture-of-Experts in Large Vision-Language Model"
_ICLR.cc/2026/Conference — Submitted to ICLR 2026_

### Official Review · Reviewer_Cafp · 2025-10-20

**Soundness:** 3
**Presentation:** 3
**Contribution:** 2
**Rating:** 6
**Confidence:** 4

**Summary:**

The authors introduce the Long-Tailed Distribution-aware Router (LTDR), a MoE routing strategy specifically designed for LVLMs. The core premise is that the standard MoE load balancing loss conflicts with the inherent long-tailed distribution of visual tokens, hindering expert specialization on salient, high-information (tailed) tokens. LTDR addresses this via two strategies: Modality-specific Distribution-aware Router (MsDaR), which removes the load balancing loss for visual tokens to increase their Routing Probability Variance (RPV), and Vision-specific Dynamic Expert Activation (VsDEA), which increases the active expert count ($k$) for tokens identified as high-RPV vision tail tokens.

**Strengths:**

1.  **Novel Modality-Specific Motivation:** The paper provides a well-motivated critique of standard MoE routing in LVLMs by highlighting the distributional mismatch between visual (long-tailed) and language (more uniform) tokens. This modality-aware design is a novel and important direction for MoE in multimodal settings.

2.  **Intuitive Two-Part Mechanism:** The proposed solution is architecturally clean, addressing both the specialization aspect (MsDaR via RPV maximization) and the learning sufficiency aspect (VsDEA via dynamic $k$) for critical tail tokens.

3.  **Clarity of Presentation:** The methodology is clearly presented with corresponding graphs and mathematical formulations.

**Weaknesses:**

1.  **Risk of Expert Collapse and Learning Stability of MsDaR:**
The removal of the explicit load balancing loss for visual tokens, aimed at increasing RPV, inherently introduces the risk of expert collapse, where a few experts become severely over-specialized and overloaded, undermining the fundamental purpose of MoE (distributed computation and diverse learning).

2.  **The Effectiveness of the Core Motivation:**
The empirical gains shown in Table 1 appear marginal, especially in smaller settings (e.g., Q-1.8B), leading to my concern of its effectiveness. Empirically, with sufficient training, standard MoE experts might naturally learn to differentiate between high-information foreground tokens (tail) and low-information background tokens (head), spontaneously increasing the RPV for critical tokens.

3.  **Experimental Scale and Generalization:**
The entire evaluation is limited to a single configuration using the LLaVA + ShareGPT dataset. MoE architectures typically realize their full benefits and stability when validated on llarge-scale pretraining or instruction-tuning datasets.

4.  **Suggestion for Future Work:**
While understandable for baseline comparison, the use of older encoders and LLMs limits the paper's immediate relevance and impact. For future work, it is strongly suggested that the method be migrated and validated using state-of-the-art vision encoders and LLMs (e.g., Siglip2 and Qwen2.5 series) to showcase the method's potential in competitive settings.

**Questions:**

1. The authors should provide a discussion and empirical evidence (e.g., expert load statistics or variance analysis over training) demonstrating the robustness of the MoE structure (vision parts) without the visual balancing loss. How is the expert load distribution maintained, and what implicit constraints prevent catastrophic expert dependency?

2. The authors should provide a direct comparison of the RPV distribution for visual tokens between a standard MoE baseline and LTDR at different training stages to demonstrate that the baseline fails to achieve the desired specialization, or that LTDR significantly accelerates this process. Marginal performance gains fail to justify the mechanical effectiveness.

3. The authors should demonstrate the scalability and generalization ability of LTDR by evaluating its performance and stability across larger-scale pretraining or SFT datasets to ensure the method holds value as a general MoE strategy.

---

> ### Author Response · Authors · 2025-11-26
> **Response to Reviewer Cafp**
>
> We sincerely thank reviewer **Cafp** for the thoughtful and constructive feedback, which will help improve our work. Our responses are as follows.
>
> > W1: Risk of expert collapse and learning stability of MsDaR.
>
> ***Response:*** As shown in ***Appendix B.1 Line 920-940 and D.1 Line 1114-1126***, removing vision load balancing does not cause significantly expert collapse compared to the baseline. In some layers, without vision load balancing even results in a reduced and more evenly distributed expert load. We attribute this to the long-tailed distribution of vision tokens. As discussed in ***Main 3.2 Line 253-254***, the abundant vision head tokens are sufficiently diverse to be distributed across experts, resulting in relatively uniform expert utilization and preventing severe expert collapse. Vision tail tokens account for only about 13% of the total and retain a reasonable probability of being routed to different experts. Therefore, catastrophic expert collapse and learning instability are not significant.
>
> > W2: The effectiveness of the core motivation.
>
> ***Response:*** While performance gains on Qwen-1.8B are modest, our method yields substantial improvements on StableLM-1.6B and achieves notable results on the recent vision-language MoE model, Molmo. This demonstrates the broad applicability of our approach across both large and small models. Even with sufficient training, standard MoE can distinguish between vision head and tail tokens. However, the relative scarcity of vision tail tokens, coupled with the constraints of the load-balancing mechanism, often prevents standard MoE from effectively learning these underrepresented features. As shown in ***W3***, even when the training data is nearly doubled, standard MoE still fails to separate vision foreground and background tokens as effectively as our method, thereby limiting its capacity for targeted learning.
>
> > W3: Experimental Scale and Generalization.
>
> ***Response:*** We conduct experiments using an expanded training dataset to provide a more intuitive comparison. Specifically, we incorporate the Open-LLaVA-NeXT training set, which adds 350K samples to the original 665K samples. Results, summarized in the table below, demonstrate that our method remains effective at larger data scales and outperforms the baseline on the full 1,021K dataset. Furthermore, the improvement in LTDR is further amplified, from 1.2% (665K) to 2.0% (1021K).
>
> **Table: Validation on Large-Scale Instruction-tuning Datasets**
> | MoE-LLaVA_StableLM | Data | GQA | ScienceQA | TextVQA | POPE | MME | MMBench | MM-Vet | Avg |
> |--------------------|------|-----|-----------|---------|------|-----|---------|--------|-----|
> | Vanilla | 665K | 60.3 | 62.6 | 50.1 | 85.7 | 1318.2 | 60.2 | 26.9 | 57.6 |
> | LTDR | 665K | 61.1 | 63.4 | 51.1 | 86.6 | 1363.5 | 60.6 | 29.9 | 58.8 |
> | Vanilla | 1021K | 61.0 | 63.0 | 51.2 | 86.7 | 1360.3 | 61.0 | 29.2 | 58.6 |
> | LTDR | 1021K | 61.4 | 65.4 | 52.5 | 87.3 | 1409.2 | 62.4 | 34.8 | 60.6 |
>
> > W4: Suggestion for future work.
>
> ***Response:*** We acknowledge the suggestion regarding the choice of encoders and LLMs, and plan to investigate more recent visual encoders and LLMs to further validate our method and explore its potential across diverse architectural foundations.
>
> > Q1: What implicit constraints prevent catastrophic expert dependency.
>
> ***Response:*** We attribute the implicit constraint that prevents catastrophic expert dependency to the long-tailed distribution of vision tokens. Please refer to ***W1*** for details.
>
> > Q2: RPV distribution at different training stages.
>
> ***Response:*** We provide updated visualizations of training progression in ***Appendix D.2 Line 1230-1233***, showing the RPV distribution at 1,500, 3,000, and 4,500 steps on TextVQA. The results indicate that, although both the baseline and our method improve mean RPV during training, our approach achieves substantially greater enhancement. This suggests that the baseline has limited capacity to effectively capture critical visual information.
>
> > Q3: The scalability and generalization ability of LTDR.
>
> ***Response:*** We demonstrate the scalability and generalization ability of LTDR by evaluating its performance and stability across the larger-scale SFT datasets to ensure the method holds value as a general MoE strategy. Please refer to ***W3*** for details.

---

### Official Review · Reviewer_eGjB · 2025-10-28

**Soundness:** 2
**Presentation:** 3
**Contribution:** 2
**Rating:** 4
**Confidence:** 4

**Summary:**

This paper proposes LTDR, which (1) eliminates load balancing for vision tokens and (2) assigns more experts to tail vision tokens. Experimental results demonstrate consistent performance improvements on both vision and vision-language tasks.

**Strengths:**

1. The motivation is clear, and the paper is easy to follow.
2. Numerical results show performance gain compare to baselines.

**Weaknesses:**

1. **Efficiency and All-to-All Communication Implementation**:
   The paper claims an efficient implementation of all-to-all communication. However, since different tokens may activate a varying number of experts under LTDR, standard efficient MoE frameworks (e.g., Deepspeed and Tutel)—which typically assume uniform expert activation—may not apply directly. Could the authors clarify how they achieve efficient all-to-all communication in this dynamic setting? Additionally, why is LTDR more computationally efficient than MoE-LLaVA despite activating more experts for tail vision tokens?

2. **Technical Novelty**:
   The core modifications—removing the load-balancing loss and increasing the number of experts for tail vision tokens—appear relatively straightforward. Could the authors elaborate on what they consider the key technical innovations of LTDR, and how it meaningfully advances beyond existing MoE or routing strategies in multimodal settings?

3. **Attribution of Performance Gains**:
   It is unclear whether the observed performance improvements stem primarily from increased model capacity (i.e., more activated parameters due to higher expert activation) rather than improved routing or representation learning. To better understand this, could the authors report the average top-K value per token and the total number of activated parameters during inference for both LTDR and baseline models?

4. **Scalability to High Vision-Token Scenarios**:
   In tasks such as video understanding, the number of vision tokens can vastly exceed that of text tokens. How does LTDR scale in such high-token regimes? Does the method incur significant computational or memory overhead, and have the authors evaluated its performance or efficiency on long-sequence or dense vision tasks?

**Questions:**

See above

---

> ### Author Response · Authors · 2025-11-26
> **Response to Reviewer eGjB**
>
> We sincerely thank reviewer **eGjB** for the thoughtful and constructive feedback, which will help improve our work. Our responses are as follows.
>
> > W1: How to achieve efficient all-to-all communication in dynamic setting.
>
> ***Response:*** Our primary focus is not on achieving efficient all-to-all communication, but on explaining why removing vision load balancing and activating additional experts for vision tail tokens results in minimal latency overhead. As described in ***Appendix B.1 Line 920-940***, this efficiency stems from MoE's all-to-all communication mechanism, where inference speed is constrained by the slowest expert (i.e., the expert processing the largest number of tokens). To validate this, we compare the expert load of MoE-LLaVA and our method on MME. The results show that our method increases the expert activation rate without significantly raising the load of the slowest expert, and even reduces it compared to the baseline, thereby improving inference efficiency.
>
> > W2: The key technical innovations of LTDR, and how it meaningfully advances beyond existing MoE or routing strategies in multimodal settings?
>
> ***Response:*** The key technical innovations of LTDR are its modality-specific, distribution-aware routing and its vision-specific dynamic expert activation. Together, these modules advance existing MoE architectures by increasing the routing probability variance of vision tokens—particularly vision tail tokens—enabling them to be processed by more specialized experts. In addition, increasing the number of activated experts for important vision tail tokens improves fault tolerance during specialized expert selection.
>
> > W3: Attribution of performance gains.
>
> ***Response:*** First, as shown in ***Main 4.3 Line 370-402 and Appendix C.2 Line 1097-1110***, our improved routing strategy alone improves performance, even without additional parameters. To further investigate, we conduct an ablation by swapping routing strategies—retaining load balancing for vision tokens while removing it for language tokens. Results in the first table below indicate that arbitrarily removing load balancing does not produce meaningful gains. Second, we also examine whether simply increasing parameters enhances performance. Adding more experts on the language side under the swapped routing setup does not affect performance, and increasing the number of activated experts for vision tail tokens to three yields no significant improvement. These findings confirm that indiscriminately increasing expert activation does not guarantee better performance and can even be detrimental. The detailed configurations for expert activation and parameter counts are provided in the second table below.  In summary, our improvements stem from two key factors: 1) Modality-specific distribution-aware routing, which allows experts to specialize according to each modality's distribution, therefore enables vision tail tokens to be processed by specialized experts. 2) Vision-specific dynamic expert activation, which enhances fault tolerance in expert selection for vision tail tokens.
>
> **Table1: Strategy-swap Ablation on Vision and Language**
> | MoE-LLaVA_StableLM | GQA | ScienceQA | TextVQA | POPE | MME | MMBench | MM-Vet | Avg |
> |--------------------|-----|-----------|---------|------|-----|---------|--------|-----|
> | Vanilla | 60.3 | 62.6 | 50.1 | 85.7 | 1318.2 | 60.2 | 26.9 | 57.6 |
> | Language |
> | + MsDaR | 60.8 | 62.0 | 50.2 | 85.9 | 1254.1 | 60.0 | 28.0 | 57.8 |
> | + MsDaR&TsDEA(a=4) | 60.1 | 61.2 | 50.4 | 86.2 | 1282.4 | 60.1 | 28.6 | 57.8 |
> | Vision |
> | + MsDaR | 61.1 | 62.3 | 51.2 | 86.6 | 1324.3 | 59.9 | 27.9 | 58.2 |
> | + MsDaR&VsDEA(a=3) | 60.7 | 62.6 | 50.8 | 86.4 | 1345.7 | 60.4 | 27.3 | 58.0 |
> | + MsDaR&VsDEA(a=4) | 61.1 | 63.4 | 51.1 | 86.6 | 1363.5 | 60.6 | 29.9 | 58.8 |
>
> **Table2: Average Top-k, Activated Parameters and Performance during Inference**
>
> | MoE-LLaVA_StableLM | Top-k | Activated Parameters (Billions) | Average Performance |
> |--------------------|-------|---------------------------------|-----------|
> | Vanilla | 2 | 4.9 | 57.6 |
> | LTDR | 2.26 | 5.2 | 58.8 |
>
> > W4: Scalability to higher vision-token scenarios.
>
> ***Response:*** Due to time constraints, we evaluate inference performance on a multi-image visual question answering task, comparing our method with the baseline in terms of both accuracy and inference speed. Specifically, we use the NLVR2 dataset, where each sample consists of two images and a textual statement. The model must determine whether the statement correctly describes the content of both images. Experimental results, presented in the table below, show that our method outperforms the baseline by +1.5% in scenarios involving a higher number of vision tokens, while maintaining nearly identical inference time and introducing no significant latency overhead.
>
> **Table: Scalability on NLVR2**
> | MoE-LLaVA-StableLM | NLVR2 | Inference Time (s) |
> |-|-|-|
> | Vanilla | 52.9 | 1121 |
> | LTDR | 54.4 | 1124 |

---

### Official Review · Reviewer_VYrS · 2025-11-01

**Soundness:** 3
**Presentation:** 3
**Contribution:** 3
**Rating:** 4
**Confidence:** 3

**Summary:**

The paper proposes LTDR, a long-tailed distribution-aware router for MoE-based LVLMs. It observes that language tokens exhibit near-uniform token-expert routing (TER), whereas vision tokens are long-tailed; standard load balancing thus misaligns with vision TER and can hinder expert specialization. LTDR consists of:
(i) MsDaR — retain load balancing for language but remove it for vision, increasing routing probability variance (RPV) and specialization.
(ii) VsDEA — identify high-RPV vision “tail” tokens and activate more experts (k→a) for them. Experiments show consistent gains across LVLMs and vision-only tasks with no significant increase in inference time.

**Strengths:**

1.	Simple, compatible design: MsDaR only alters the balancing objective by modality; VsDEA uses a mean-RPV threshold and k→a activation. Easy to integrate into existing MoE stacks.
2.	Broad, consistent improvements: Steady gains across MoE-LLaVA and Molmo; vision-only GMoE also benefits (e.g., ~+1.0% Avg).
3.	Latency and efficiency: Inference time remains essentially unchanged; the all-to-all bottleneck explanation is plausible and supported by expert-load analysis.

**Weaknesses:**

1. The paper describes visual tokens as "long-tailed" based on RPV histograms (Figure 1(b)), but lacks statistical tests (e.g., tail index, power-law fitting) to rigorously support this claim.
RPV is influenced by router parameters (gate temperature, regularization), so its "tail" may not reflect semantic or information-based long-tailedness.

2. The choice of base models is narrow, limiting generalizability. Descriptions like "Qwen-1.8B" are too vague; specific model details are needed.

3. No comparison with simpler methods (e.g., temperature/entropy scaling) to sharpen visual token routing, which could also increase RPV without disabling load balancing.

4. The mean-RPV threshold (Equation 11) lacks ablation studies (e.g., generalizability across datasets/models). Only fixed proportion thresholds (10/15/20%) are tested (Appendix B.5), with no deeper sensitivity analysis.

5. MsDaR removes load balancing for visual tokens (Equation 10), but risks like expert overloading or training instability under large-scale K (e.g., 64) or extreme data distributions are not quantified.

6. Main results show small improvements (+0.4 to +1.2, Table 1), but significance tests are only in Appendix C.2 (Table 18).
Confidence intervals and variance should be included in the main text.

7. The impact of reducing (not removing) load balancing or adding constraints (e.g., temperature/entropy) is not studied.
How do methods like temperature annealing or entropy regularization compare to VsDEA?

8. The link between RPV and token informativeness relies on visualizations/ablations; stronger statistical or interpretability evidence is needed.

9.  VsDEA’s choice of activated experts (a) is empirical; systematic tuning or adaptive policies would improve credibility.

10. Benefits under near-uniform token distributions are unclear; synthetic studies could help define boundary conditions.

11. Batch-wise mean-RPV thresholds may misclassify non-critical patches as "tails," diverting experts from important regions. A learnable joint mechanism is suggested.

**Questions:**

1.	RPV ↔ informativeness: Can you quantify correlation between high-RPV vision tokens and downstream performance or gradient-based relevance (e.g., leave-one-token-out, attribution metrics)?
2.	Adaptive a: Could a be chosen via RPV quantiles or a compute-budgeted policy? Any results across different K/k?
3.	Training stability: Does removing vision load balancing cause early expert skew or collapse? Is entropy regularization or warm-up needed?

---

> ### Author Response · Authors · 2025-11-26
> **Response to Reviewer VYrS (1/3)**
>
> We sincerely thank reviewer **VYrS** for the thoughtful and constructive feedback, which will help improve our work. Our responses are as follows.
>
> > W1: Lacking statistical tests to support "visual tokens are long-tailed".
>
> ***Response:*** Under fixed router parameters (gate temperature and regularization), we compare the vision token RPV distributions across different routers and backbones (***Appendix D.2 Line 1234-1241***). These distributions exhibit a clear long-tailed pattern: they are highly asymmetric, with the majority of probabilities concentrated in the head region, while the remaining probabilities form an extended tail where sample frequency decreases sharply toward higher values.
>
> > W2: The choice of base models is narrow, and specific model details are needed.
>
> ***Response:*** We select two representative vision-language MoE models: MoE-LLaVA, the earliest open-source and widely studied model with three different backbones in this category, and Molmo, the most recent state-of-the-art vision-language MoE model (***Main 4.2 Lines 319–323***). In addition, we evaluate the generalization of our method on the visual MoE model GMoE (***Main 4.2 Lines 365–368***). All these models differ in scale and architecture. For Qwen-1.8B, we extend the Qwen-1.8B language model by integrating a visual projection module (CLIP) and replicating the MLP parameters in the transformer layers to construct multiple experts with identical initialization.
>
> > W3: No comparison with simpler methods to sharpen vision token routing.
>
> ***Response:*** To test the sharpening of vision token routing, we compare our method with several simpler routing strategies, including temperature scaling, entropy regularization, and heuristic sharpening. The results are summarized in the table below. Temperature-based: Gumbel-Softmax with a temperature of 0.7. Entropy-based: Standard entropy regularization to encourage more confident routing. Heuristic-based: Constraint on the variance of vision token routing. Simply sharpening vision token routing does not yield significant improvements. We hypothesize that this is due to the distinct routing behaviors of head and tail vision tokens: tail tokens are relatively few, so increasing vision token routing can help them reach specific experts more effectively; in contrast, head tokens are abundant and sufficiently learn from various experts, so increasing vision token routing may actually hinder the learning of tail tokens.
>
> **Table: Comparison with Simpler Sharpening Vision Token Routing Methods**
> | MoE-LLaVA-StableLM | GQA | ScienceQA | TextVQA | POPE | MME | MMBench | MM-Vet | Avg |
> |--------------------|-----|-----------|---------|------|-----|---------|--------|-----|
> | Vanilla | 60.3 | 62.6 | 50.1 | 85.7 | 1318.2 | 60.2 | 26.9 | 57.6 |
> | Temperature-based | 60.9 | 62.8 | 50.2 | 86.0 | 1302.5 | 60.1 | 26.8 | 57.8 |
> | Entropy-based | 61.1 | 62.4 | 50.8 | 86.2 | 1330.7 | 60.0 | 27.2 | 57.9 |
> | Heuristic-based | 61.1 | 62.5 | 50.3 | 85.8 | 1299.6 | 60.2 | 26.6 | 57.7 |
> | LTDR | 61.1 | 63.4 | 51.1 | 86.6 | 1363.5 | 60.6 | 29.9 | 58.8 |
>
> > W4: The mean-RPV threshold lacks ablation studies (e.g., generalizability across datasets/models).
>
> ***Response:*** We compare the mean RPV using thresholds of 10%, 15%, and 20%, as the observed mean RPV (13%) lies between these values. This allows us to visually assess how small variations in the threshold affect performance. We have evaluated these fixed-proportion thresholds on the MoE-LLaVA with StableLM-1.6B model across multiple test datasets in ***Appendix B.5 Line 1064-1079***. Here, we further present generalizability experiments on the MoE-LLaVA with Phi-2-2.7B model. The results in the table below also show that mean-RPV has a consistent advantage over the comparison threshold. We also compare mean-RPV with the learnable strategy presented in ***W9***; however, the tuning method yields performance that falls between that of the original model and the LTDR-based model.
>
> **Table: Generalizability across Different Models**
> | Model | GQA | ScienceQA | TextVQA | POPE | MME | MMBench | MM-Vet | Avg |
> |-------|-----|-----------|---------|------|-----|---------|--------|-----|
> | Phi-2 with 10% | 62.1 | 68.3 | 51.6 | 86.5 | 1406.3 | 65.6 | 33.7 | 61.3 |
> | Phi-2 with 15% | 62.0 | 68.4 | 51.8 | 86.6 | 1423.1 | 66.3 | 34.0 | 61.5 |
> | Phi-2 with 20% | 61.3 | 67.8 | 51.3 | 86.2 | 1414.5 | 66.5 | 34.1 | 61.2 |
> | Phi-2 with mean-RPV | 62.2 | 68.5 | 52.0 | 86.7 | 1440.8 | 66.7 | 34.0 | 61.7 |

---

> ### Author Response · Authors · 2025-11-26
> **Response to Reviewer VYrS (2/3)**
>
> > W5: Expert overloading or training instability under large-scale K or extreme data distributions are not quantified.
>
> ***Response:*** Molmo represents the latest vision-language MoE model under large-scale K (64) configuration. Our method achieves a measurable performance improvement (***Main 4.2 Line 319-354***) on this architecture while introducing no significant inference latency (the table below) compared to the vanilla model, demonstrating both the efficiency and stability of our approach.
>
> **Table: Molmo Inference-time Latency**
> | Method | GPU | ChartQA | DocVQA | AI2D | VQA | AndroidControl | CountBenchQA | Avg (s) |
> |--------|-----|---------|--------|------|-----|----------------|--------------|---------|
> | Molmo | A800-80G | 184 | 200 | 190 | 196 | 89 | 705 | 261 |
> | +LTDR | A800-80G | 187 | 203 | 194 | 200 | 90 | 716 | 265 |
>
> > W6: Confidence intervals and variances should be included in the main text.
>
> ***Response:*** Thank you for the suggestion. We have incorporated this content into ***Main 4.5 Line 474-477***.
>
> > W7: The impact of reducing load balancing or adding constraints is not studied.
>
> ***Response:*** Please refer to ***W3*** for the impact of adding sharpening vision token routing constraints (e.g., entropy regularization). Here, we analyze the effect of reducing load balancing, with results presented in the table below. Specifically, we decrease the vision-side load balancing coefficient from 0.01 to 0.001. The results indicate that this adjustment is less effective than removing the load balancing.
>
> **Table: Impact of Reducing Load Balancing**
> | MoE-LLaVA-StableLM | GQA | ScienceQA | TextVQA | POPE | MME | MMBench | MM-Vet | Avg |
> |-|-|-|-|-|-|---------|--------|-----|
> | Vanilla | 60.3 | 62.6 | 50.1 | 85.7 | 1318.2 | 60.2 | 26.9 | 57.6 |
> | Reducing-based | 60.2 | 62.4 | 50.9 | 86.3 | 1315.7 | 59.1 | 28.4 |	57.8 |
> | LTDR | 61.1 | 63.4 | 51.1 | 86.6 | 1363.5 | 60.6 | 29.9 | 58.8 |
>
> > W8: The link between RPV and token informativeness needs stronger statistical or interpretability evidence.
>
> ***Response:*** We evaluate the link between RPV and token informativeness from two aspects: 1) Performance-based analysis (interpretability). Higher model performance suggests that the corresponding tokens carry richer information. As described in ***Main 4.4, lines 415–425***, we sort tokens by RPV in descending order and classify those above the mean RPV as vision tail tokens (13%) and the remainder as vision head tokens (87%). Although vision head tokens are nearly $7 \times$ more numerous than tail tokens, applying the VsDEA strategy to head tokens yields substantially lower performance compared to applying it to tail tokens. This performance gap indicates that high-RPV vision tail tokens encode more informative content. 2) Statistics-based analysis (statistical). A token group with a higher mean L2 norm of its vector representations reflects richer underlying information. We compare the mean L2 norm of the top 13% vision token vectors (vision tail tokens) with those of the vectors ranked between the top 13%-26% and the top 26%-39% by RPV. As shown in the table below, the first group exhibits a higher mean L2 norm than the latter two, providing further evidence that high-RPV vision tail tokens carry richer information.
>
> **Table: Statistics-based analysis**
> |  | mean L2 norm |
> |-|-|
> | Top-13% | 0.3158 |
> | Top-13% - Top-26% | 0.2124 |
> | Top-26% - Top-39% | 0.1475 |
>
> > W9: VsDEA’s choice of activated experts is empirical, systematic tuning or adaptive policies would improve credibility.
>
> ***Response:*** Please refer to ***Q2*** for the adaptive policy. Here, we analyze the systematic tuning policy. Specifically, we introduce a learnable RPV threshold to identify vision tail tokens. Experimental results are summarized in the table below. The performance of the tuning method falls between that of the original model and the LTDR-based model.
>
> **Table: Systematic Tuning Policy**
> | MoE-LLaVA-StableLM | GQA | ScienceQA | TextVQA | POPE | MME | MMBench | MM-Vet | Avg |
> |-|-|-|-|-|-|-|-|-|
> | Vanilla | 60.3 | 62.6 | 50.1 | 85.7 | 1318.2 | 60.2 | 26.9 | 57.6 |
> | Tuning-based | 61.0 | 62.1 | 50.6 | 86.3 | 1322.3 | 59.9 | 28.9 | 58.1 |
> | LTDR | 61.1 | 63.4 | 51.1 | 86.6 | 1363.5 | 60.6 | 29.9 | 58.8 |
>
> > W10: Benefits under near-uniform token distributions are unclear.
>
> ***Response:*** We investigate a multimodal MoE architecture for vision–language tasks. Owing to the inherently non-uniform token distribution in images, it is difficult to evaluate the benefits of our approach under near-uniform token distributions. Nevertheless, we find that although our method is designed to improve vision-token processing, it does not degrade the model’s ability to handle near-uniform language tokens. As shown in ***Main 4.2 Line 319-346***, datasets such as SQA contain language data with near-uniform token distributions, yet our model preserves performance on these tasks.

---

> ### Author Response · Authors · 2025-11-26
> **Response to Reviewer VYrS (3/3)**
>
> > W11: Batch-wise mean-RPV thresholds may misclassify non-critical patches as tails. A learnable joint mechanism is suggested.
>
> ***Response:*** We investigate a learnable threshold learning mechanism for identifying vision tail tokens, and details are provided in ***W9***. However, the threshold learned in this manner does not generalize well to downstream tasks. As indicated by our experimental results, this learnable approach performs at an intermediate level between the original model and the LTDR-based model. In future work, we aim to more precisely delineate the RPV boundary between vision head and tail tokens, which we expect will improve the generalizability of our method across diverse training datasets and model architectures.
>
> > Q1: The quantify correlation between high-RPV vision tokens and downstream performance.
>
> ***Response:*** We relate the correlation between high-RPV vision tokens and downstream performance to the link between RPV and token informativeness. Please refer to ***W8*** for details. Moreover, the RPV distribution in ***Appendix D.2 Line 1225-1241***, together with the performance results in ***Main 4.2 Line 319-368***, further demonstrates a positive correlation between high-RPV vision tokens and downstream task performance.
>
> > Q2: Adaptive $a$: Could $a$ be chosen via RPV quantiles or a compute-budgeted policy?
>
> ***Response:*** We have studied the RPV quantiles policy in ***Appendix B.5 Line 1064-1079***. Here, we experiment with a Top-P compute-budgeted policy that fixes the number of tokens per expert while dynamically selecting experts for each token. Specifically, expert activation is determined by token score ranking and expert capacity, where expert capacity is defined as $num_{\text{token}} / num_{\text{experts}} \times 3$. As shown in the table below, this approach hinders performance improvement, indicating a more complex underlying issue that warrants further investigation.
>
> **Table: Top-P Compute-budgeted Policy**
> | MoE-LLaVA-StableLM | GQA | ScienceQA | TextVQA | POPE | MME | MMBench | MM-Vet | Avg |
> |--------------------|-----|-----------|---------|------|-----|---------|--------|-----|
> | LTDR+Top-P | 60.8 | 62.1 | 51.1 | 86.8 | 1332.8 | 60.2 | 28.9 | 58.3 |
> | LTDR+Top-a | 61.1 | 63.4 | 51.1 | 86.6 | 1363.5 | 60.6 | 29.9 | 58.8 |
>
> > Q3: Training stability: Does removing vision load balancing cause early expert skew or collapse?
>
> ***Response:*** As shown in ***Appendix B.1 Line 920-940 and D.1 Line 1114-1126***, removing vision load balancing does not cause significantly expert collapse compared to the baseline. In some layers, without vision load balancing even results in a reduced and more evenly distributed expert load. We attribute this to the long-tailed distribution of vision tokens. As discussed in ***Main 3.2 Line 253-254***, the abundant vision head tokens are sufficiently diverse to be distributed across experts, resulting in relatively uniform expert utilization and preventing severe expert collapse. Vision tail tokens account for only about 13% of the total and retain a reasonable probability of being routed to different experts. Therefore, catastrophic expert collapse is not significant.

---

### Official Review · Reviewer_ZRpW · 2025-11-01

**Soundness:** 3
**Presentation:** 3
**Contribution:** 3
**Rating:** 6
**Confidence:** 3

**Summary:**

The paper revolves around the contradiction in MoE routing (TER) within LVLMs between “load balancing preferring uniform distribution” and “inconsistency in modality distributions.” Through multiple experiments and visualizations, the authors observe that the TER probability variance distribution for language tokens is closer to uniform, whereas for visual tokens it exhibits a clear long-tail pattern. Under these conditions, enforcing load balancing on visual tokens disrupts the specialization of the “tail” visual tokens that are rare yet critical. Based on this, the authors propose LTDR, which includes MsDaR, applying different balancing constraints to language and vision, and VsDEA, which uses routing probability variance (RPV) to identify “tail” visual tokens online and activate more experts for them.

The core idea is to retain load balancing only on the language side and remove it for the visual side to increase RPV; a dynamic threshold is used to route tail tokens to Top-$a$ experts instead of Top-$k$. The authors validate this on MoE-LLaVA, Molmo, and GMoE, achieving average gains of about +1.2%, +2.0%, and +1.0%, respectively, and provide ablations, routing comparisons, runtime and memory statistics, and visualization analyses. It should be noted that the conclusion “language nearly uniform, vision long-tailed” is primarily supported by the examples and visualizations in the paper, and its applicability still depends on the model and router setup; it cannot yet be regarded as a cross-architecture law.

**Strengths:**

The problem setup is practice-oriented: when the MoE balancing term favors “uniformity” but visual tokens are “long-tailed,” rare yet key information can indeed be diluted. LTDR’s two modifications are both minimal: MsDaR merely limits the balancing term to language sequences, and VsDEA activates more experts for a small number of high-RPV visual tokens without altering sequence length—effectively an “oversampling-like” opportunity expansion. The authors provide extensive comparisons, ablations, and cost analyses, including runtime, memory, expert load distribution, and random-seed intervals, making the method reproducible and practically valuable.

**Weaknesses:**

The “language near-uniform, vision long-tailed” characterization currently depends on the chosen models and visual tokenizer (Appendix D.2 notes 576 CLIP tokens per image) as well as specific router/Top-k settings. Although extensive experiments are conducted on MoE-LLaVA, Molmo, and GMoE, it remains unclear whether this distribution pattern holds when changing router types (e.g., dynamic expert counts, segmented softmax, top-1 vs top-2), expert scales and layers, or visual tokenizers. A systematic validation is lacking. I suggest adding cross-router/backbone RPV distribution comparisons and variance analyses to avoid over-generalizing empirical observations.

Moreover, the implementation of VsDEA Top-a is somewhat heuristic: Eq. (12) does not specify whether weights are renormalized or if temperature/thresholding is used to suppress noise from low-probability experts; inference-time latency and all-to-all waiting for a=12 versus k=8 on Molmo are not quantified (only MoE-LLaVA reports timing). In addition, slight degradation appears on the POPE Adversarial subset, suggesting that “activating more experts for tail tokens” may introduce bias toward overconfidence on adversarial negatives—this deserves further analysis and mitigation.

**Questions:**

I would especially like the authors to perform a “strategy-swap” ablation: remove load balancing on the language side while keeping it on the vision side, and examine the corresponding MsDaR and VsDEA variants. Observing whether de-balancing language routing harms its originally uniform distribution or introduces expert bias in instruction-alignment tasks would directly support (or challenge) the claim that “modality-specific strategies are necessary.” Conversely, if performance improves after swapping, it would significantly refine the boundary conditions of the conclusion.

Additionally, I hope to see two implementation and measurement clarifications:

Details on normalization, clipping, and temperature settings in Top-a, and their impact on gradients and communication costs.

Broader RPV distribution and performance comparisons across routers/backbones, especially including dynamic routing and top-1 gating.

Finally, regarding POPE Adversarial degradation, could VsDEA incorporate confidence- or consistency-based gating, or be combined with methods like STGC (gradient conflict mitigation) to suppress overconfidence when seeing partial visual evidence? Addressing these points could further raise my overall evaluation.

---

> ### Author Response · Authors · 2025-11-26
> **Response to Reviewer ZRpW (1/2)**
>
> We sincerely thank reviewer **ZRpW** for the thoughtful and constructive feedback, which will help improve our work. Our responses are as follows.
>
> > W1: Adding cross router/backbone RPV distribution comparisons and variance analyses.
>
> ***Response:*** We have updated the cross-router/backbone RPV distribution in ***Appendix D.2 Line 1234-1241***. Cross-Router: In the Top-Dynamic setting, expert activation is determined by token score ranking and expert capacity, where expert capacity is defined as $num_{\text{token}} / num_{\text{experts}} \times 3$. The results indicate that the mean RPV of the vision tail under Top-1 is significantly lower than in other configurations, suggesting that increasing the number of parameters can enhance vision tail token learning. However, despite Top-Dynamic having the largest number of parameters, it does not outperform Top-a in terms of vision tail RPV, implying that simply increasing parameter count does not guarantee performance improvement. Cross-Backbone: The choice of backbone has a notable impact on mean RPV. Among the evaluated backbones, Phi-2 achieves the highest mean RPV, followed by StableLM, with Qwen yielding the lowest. The performance of the models aligns with this ranking.
>
> > W2: Details on VsDEA implementation and Molmo inference-time latency, and an analysis on the POPE adversarial subset.
>
> ***Response:*** VsDEA Implementation: We have updated the renormalization procedure in ***Eq.12*** and ***Main 3.3 Line 296***. Our implementation, however, adheres to the MoE frameworks of the original models and does not specifically suppress noise from low-probability experts by using temperature/thresholding. Molmo Inference-Time Latency: As shown in the table below, there is no significant difference in inference latency between the configurations with $a=12$ (LTDR) and $k=8$ (Molmo). Analysis of the POPE: We attribute this to the recognition of vision tail tokens. As detailed in ***Appendix E Line 1269-1276***, such tokens are identified at the batch level, which may be constrained by potential false positives. Addressing this limitation constitutes an important direction for future research.
>
> **Table: Molmo Inference-time Latency on the A800-80G GPU**
> | Method | GPU | ChartQA | DocVQA | AI2D | VQA | AndroidControl | CountBenchQA | Avg (s) |
> |--------|-----|---------|--------|------|-----|----------------|--------------|---------|
> | Molmo | A800-80G | 184 | 200 | 190 | 196 | 89 | 705 | 261 |
> | +LTDR | A800-80G | 187 | 203 | 194 | 200 | 90 | 716 | 265 |
>
> > Q1: Strategy-swap ablation on the language side while keeping it on the vision side.
>
> ***Response:*** We perform a strategy-swap ablation on the language side while keeping it on the vision side. The results, summarized in the table below, show that removing the load balancing loss on the language side (language+MsDAR) leads to fluctuating performance, with only a 0.2% average improvement compared to the vanilla model. Additionally, introducing text-specific dynamic expert activation (TsDEA) does not result in consistent performance gains. In contrast, eliminating the load balancing loss on the vision side (vision+MsDAR) produces stable improvements, and further incorporating vision-specific dynamic expert activation (VsDEA) enhances performance, achieving an average improvement of 1.2%.
>
> **Table: Strategy-swap Ablation on Vision and Language**
> | MoE-LLaVA_StableLM | GQA | ScienceQA | TextVQA | POPE | MME | MMBench | MM-Vet | Avg |
> |--------------------|-----|-----------|---------|------|-----|---------|--------|-----|
> | Vanilla | 60.3 | 62.6 | 50.1 | 85.7 | 1318.2 | 60.2 | 26.9 | 57.6 |
> | Language |
> | + MsDaR | 60.8 | 62.0 | 50.2 | 85.9 | 1254.1 | 60.0 | 28.0 | 57.8 |
> | + MsDaR&TsDEA | 60.1 | 61.2 | 50.4 | 86.2 | 1282.4 | 60.1 | 28.6 | 57.8 |
> | Vision |
> | + MsDaR | 61.1 | 62.3 | 51.2 | 86.6 | 1324.3 | 59.9 | 27.9 | 58.2 |
> | + MsDaR&VsDEA | 61.1 | 63.4 | 51.1 | 86.6 | 1363.5 | 60.6 | 29.9 | 58.8 |
>
> > Q2: Details on normalization, clipping, and temperature settings in Top-a, and their impact on gradients and communication costs.
>
> ***Response:*** We perform renormalization after expert selection, without introducing any additional clipping or temperature settings in Top-a. Please refer to ***W2*** for details. Nevertheless, we believe that modifying clipping or temperature does affect both gradients and communication costs. For example, clipping can reduce communication costs by suppressing certain tokens, but it also causes gradients to descend more slowly or become unstable due to reduced information flow, leading to performance degradation (as demonstrated in ***Q3*** by the performance differences between Vanilla+Top-1 and Vanilla+Top-2). Similarly, adjusting the temperature may lead to comparable effects.

---

> ### Author Response · Authors · 2025-11-26
> **Response to Reviewer ZRpW (2/2)**
>
> > Q3: Broader RPV distribution and performance comparisons across routers/backbones.
>
> ***Response:*** Please refer to ***W1*** for the RPV distribution across routers/backbones, and to ***Main 4.2 Line 317-368*** for performance comparisons across backbones. Here, we report the cross-router performance results, summarized in the table below. LTDR with the Top-1 achieves a 0.9% improvement over the vanilla model. Although dynamic routers provide additional gains, their performance remains below that of LTDR combined with Top-a.
>
> **Table: Cross-Router Performance**
> | MoE-LLaVA-StableLM | GQA | ScienceQA | TextVQA | POPE | MME | MMBench | MM-Vet | Avg |
> |--------------------|-----|-----------|---------|------|-----|---------|--------|-----|
> | Vanilla+Top-1 (k=1) | 58.6 | 55.8 | 45.0 | 85.2 | 1245.3 | 56.2 | 27.2 | 54.7 |
> | Vanilla+Top-2 (k=2) | 60.3 | 62.6 | 50.1 | 85.7 | 1318.2 | 60.2 | 26.9 | 57.6 |
> | LTDR+Top-1 (k=1) | 59.7 | 58.2 | 45.6 | 85.8 | 1302.9 | 57.1 | 27.0 | 55.6 |
> | LTDR+Dynamic (avg k=3) | 60.8 | 62.1 | 51.1 | 86.8 | 1332.8 | 60.2 | 28.9 | 58.3 |
> | LTDR+Top-a (avg k=2.26) | 61.1 | 63.4 | 51.1 | 86.6 | 1363.5 | 60.6 | 29.9 | 58.8 |
>
> > Q4: Can VsDEA incorporate a confidence- or consistency-based gating mechanism to suppress overconfidence at the vision side?
>
> ***Response:*** We acknowledge the feasibility of the suggested approach for suppressing overconfidence on the vision side. However, our gating mechanism is designed to allow vision tail tokens to specialize in specific experts, whereas confidence- or consistency-based gating seeks to reduce conflicts among tokens assigned to the same expert. Their objectives may conflict in certain scenarios, making it challenging to balance the two strategies. We regard this as a promising direction for future research.

---

### Author Response · Authors · 2025-12-04
**List of Revisions**

We sincerely appreciate your thoughtful and constructive feedback. We attach great importance to every comment and suggestion. According to the reviewers' suggestions, we have submitted a revised version of the paper. The main revisions are as follows:

-In **Eq.12**, we have refined the ***Renormalization Procedure*** to address the concerns of implementation details raised by ZRpW.

-In **Section 4.5**, we have moved ***Confidence Intervals and Variance*** to the main text in response to the suggestion raised by VYrS.

-In **Appendix D.2**, we have provided ***Training Stages RPV***, ***Cross-router RPV***, and ***Cross-backbone RPV*** to address the concerns of distribution comparisons raised by ZRpW, VYrS and Cafp.

-In **Appendix F.1**, we have provided ***Molmo Inference-time Latency*** to address the concerns of latency raised by ZRpW and VYrS.

-In **Appendix F.2**, we have provided ***Ablation on Vision and Language*** to address the concerns of strategy-swap ablation raised by ZRpW and eGjB.

-In **Appendix F.3**, we have provided ***Cross-router Performance*** to address the concerns of cross-router performance raised by ZRpW.

-In **Appendix F.4**, we have provided ***Comparison with Simpler Sharpening Vision Token Routing Methods*** to address the concerns of routing comparisons raised by VYrS.

-In **Appendix F.5**, we have provided ***Generalizability across Models*** and ***Comparison with Tuning Policy*** to address the concerns of generalizability and tuning comparison raised by VYrS.

-In **Appendix F.6**, we have provided ***Impact of Reducing Load Balancing*** to address the concerns of reducing balancing comparison raised by VYrS.

-In **Appendix F.7**, we have provided ***The Link between RPV and Token Informativeness*** to address the concerns of statistical evidenc raised by VYrS.

-In **Appendix F.8**, we have provided ***Average Top-k, Activated Parameters and Performance during Inference*** to address the concerns of activated parameters raised by eGjB.

-In **Appendix F.9**, we have provided ***Scalability to Higher Vision-token Scenarios*** to address the concerns of scenario scalability raised by eGjB.

-In **Appendix F.10**, we have provided ***Experimental Scale and Generalization*** to address the concerns of datset generalization raised by Cafp.

---

### Author Response · Authors · 2025-12-04
**Summary of the Discussion Phase and Paper Revisions**

Dear Area Chair,

We sincerely thank you and the reviewers for the time and insightful assessment of our work. We highly appreciate the constructive feedback, which has greatly strengthened the paper. Below, we provide a summary of the consensus on the paper’s strengths and the revisions made in response.

---

First, we appreciate that all reviewers acknowledge the contribution and novelty of this work:
* **Motivation:** The reviewers acknowledge the clarity and accessibility of the motivation (eGjB, Cafp), and further recognize the novelty (Cafp) and practical value (ZRpW) of the work.
* **Method:** The proposed method is well received by the reviewers, who regard it as simple and practical while also exhibiting good compatibility and strong performance (ZRpW, VYrS, Cafp).
* **Performance:** Having approved the method employed in this work, the reviewers further affirm its efficiency and generalizability (ZRpW, VYrS, eGjB).
* **Reproducibility:** Drawing on the extensive comparative and ablation experiments as well as the cost analysis, the reviewers consider the method to be reproducible and reliable (ZRpW, VYrS, eGjB).

---

Second, in response to the weaknesses and questions raised during review and discussion, we have made the following clarifications and additions:

**Revisions to the paper:**

***1. Comparative Experiments (ZRpW, VYrS, Cafp)***

* Cross-step/router/backbone: Figures 8–10 present cross-step, cross-router, and cross-backbone RPV distributions, and Table 22 shows cross-router performance validating LTDR (ZRpW, VYrS, Cafp).
* Strategy-swap ablation: Table 21 provides the strategy-swap ablation for vision and language (ZRpW, VYrS).
* Sharpening vision token routing: Tables 23 and 26 compare sharpening-based routing methods to validate MsDaR (VYrS).
* Threshold: Tables 22 and 25 compare compute-budgeted and learnable thresholds (VYrS).

***2. Generalization Experiments (VYrS, eGjB, Cafp)***

* Threshold generalizability across models: Table 24 presents generalizability experiments on MoE-LLaVA with the Phi-2-2.7B model to address concerns regarding threshold generalization (VYrS).
* Scalability to high vision-token scenarios: Table 29 evaluates scalability under higher vision-token settings, where LTDR outperforms the baseline with almost no increase in inference time (eGjB).
* Validation on large-scale instruction-tuning datasets: Table 30 reports results on a larger instruction-tuning dataset, where the substantial performance gains over the baseline demonstrate LTDR’s strong generalization ability (Cafp).

***3. Statistical Analysis (ZRpW, VYrS, eGjB)***

* Molmo inference-time latency: Table 20 analyzes Molmo’s inference-time latency, demonstrating the efficiency of LTDR (ZRpW, VYrS).
* Statistical evidence linking RPV and token informativeness: Table 27 compares the mean L2 norm across token groups, showing that tokens with higher RPV carry more information (VYrS).
* Activated parameters: Table 28 reports the average number of activated experts and parameters, confirming that LTDR does not rely on increased computational overhead (eGjB).

***4. Experimental Details (ZRpW, VYrS)***

VsDEA implementation: Eq.12 has been refined to address the concerns regarding implementation details (ZRpW).

**Discussion phase:** We submitted our response and revised manuscript on November 27. Unfortunately, due to the early conclusion of the discussion period, we did not receive any response or further feedback from the reviewers. We have made every effort to incorporate their feedback comprehensively into the revised manuscript. We believe the additional experiments and analyses listed above effectively address the reviewers' concerns regarding comparison, generalization, reproducibility, and model design.

---

Thank you for your time and consideration. We hope this summary assists in your decision-making process.

Sincerely,

Authors of Paper #13490

---

### Meta-Review · Area_Chair_R6sS · 2026-01-06

**Summary:**

The paper makes an interesting obersvations, in MOE-based models, language tokens exhibit near-uniform token-to-expert routing (TER), whilst vision tokens follow a long-tailed distribution instead. Therefore, standard load balancing of tokens to experts can hinder performance for the vision modality. Following this, the authors make two main contributions 1) MsDaR -- They retain load balancing for language tokens, but do not do so for vision, to increase the specialisation for vision tokens. 2) VsDEA -- Identify vision "tail" tokens with a high variance and activate more experts for these tokens.

Reviewers appreciated the overall motivation of the paper, and the authors analysis to identify this mismatch in token-distributions. Moreover, the proposed method is practical too, as it does not add much overhead, and it can increase results too.

However, the actual improvement from the method is unclear, as standard deviations are not reported for most experiments. During the rebuttal, the authors added Table 10 to the paper, but the standard deviations appear suspiciously low (and a standard deviation of 0 is even reported for 1 dataset). Re-training the model, but with a different random seed would surely introduce much more variance than this. It appears that the authors have only changed the seed during inference time? This does not answer the question of how statistically signficant the improvements actually are, given that throughout the paper, the improvement over the baseline is usually below 1 point.

Other concerns included the fact that the imbalanced visual token distribution is an empirical observation, and may not hold for more recent MoE models. Unfortunately the authors have not tested their method on the most recent VLMs, like Qwen 3 VL or InternVL.

Therefore, on the balance, the final decision is to reject this paper. Authors are encouraged to revise the paper according to the reviewer feedback and to resubmit to another venue.

**Reviewer Concerns:**

The main weaknesses mentioned by the reviewers, and not addressed sufficiently during the rebuttal include

- "Main results show small improvements (+0.4 to +1.2, Table 1), but significance tests are only in Appendix C.2 (Table 18). Confidence intervals and variance should be included in the main text."
- "The “language near-uniform, vision long-tailed” characterization currently depends on the chosen models and visual tokenizer (Appendix D.2 notes 576 CLIP tokens per image) as well as specific router/Top-k settings"
- "The choice of base models is narrow, limiting generalizability"
- "The entire evaluation is limited to a single configuration using the LLaVA + ShareGPT dataset. MoE architectures typically realize their full benefits and stability when validated on llarge-scale pretraining or instruction-tuning datasets."
- "In tasks such as video understanding, the number of vision tokens can vastly exceed that of text tokens. How does LTDR scale in such high-token regimes?"

Although other concerns have been addressed, the AC considers the aforementioned points significant.

**Reviewer Scores:**

Reviewers VYrS and eGjB, who initially rated weak reject (4) would probably have retained their scores.

Similarly, Reviewers Cafp and ZRpW, who initially rated weak accept (6) would probably have retained their scores.

---

### Decision · Program_Chairs · 2026-01-26

Reject